# Exploring the potential of structure-based deep learning approaches for T cell receptor design

Helder V. Ribeiro-Filho[1‡]*, Gabriel E. Jara[1‡], João V. S. Guerra[1,2], Melyssa Cheung[3,4], Nathaniel R. Felbinger[3,5], José G. C. Pereira[1], Brian G. Pierce[3,5]*, Paulo S. Lopes-de-Oliveira[1,2]*

**1** Brazilian Biosciences National Laboratory, Brazilian Center for Research in Energy and Materials, Campinas, São Paulo, Brazil, **2** Graduate Program in Pharmaceutical Sciences, Faculty of Pharmaceutical Sciences, University of Campinas, Campinas, São Paulo, Brazil, **3** Institute for Bioscience and Biotechnology Research, University of Maryland, Rockville, Maryland, United States of America, **4** Department of Chemistry and Biochemistry, University of Maryland, College Park, Maryland, United States of America, **5** Department of Cell Biology and Molecular Genetics, University of Maryland, College Park, Maryland, United States of America

‡ These authors joint first authorship on this work.
* helder.ribeiro@lnbio.cnpem.br (HVRF); pierce@umd.edu (BGP); paulo.oliveira@lnbio.cnpem.br (PSLO)

**Data Availability Statement:** All raw data and results related to this paper are available on Zenodo at link https://doi.org/10.5281/zenodo.11086017).

## Abstract

Deep learning methods, trained on the increasing set of available protein 3D structures and sequences, have substantially impacted the protein modeling and design field. These advancements have facilitated the creation of novel proteins, or the optimization of existing ones designed for specific functions, such as binding a target protein. Despite the demonstrated potential of such approaches in designing general protein binders, their application in designing immunotherapeutics remains relatively underexplored. A relevant application is the design of T cell receptors (TCRs). Given the crucial role of T cells in mediating immune responses, redirecting these cells to tumor or infected target cells through the engineering of TCRs has shown promising results in treating diseases, especially cancer. However, the computational design of TCR interactions presents challenges for current physics-based methods, particularly due to the unique natural characteristics of these interfaces, such as low affinity and cross-reactivity. For this reason, in this study, we explored the potential of two structure-based deep learning protein design methods, ProteinMPNN and ESM-IF1, in designing fixed-backbone TCRs for binding target antigenic peptides presented by the MHC through different design scenarios. To evaluate TCR designs, we employed a comprehensive set of sequence- and structure-based metrics, highlighting the benefits of these methods in comparison to classical physics-based design methods and identifying deficiencies for improvement.

## Author summary

Redirecting T cells to recognize and combat infected or altered cells through the engineering of T cell receptors (TCR) has shown promising results, particularly in immunotherapy

Additionally, other relevant data can be found in the main manuscript and the Supplementary Materials.

**Funding:** This work was supported by the grant #2022/04260-6, São Paulo Research Foundation (FAPESP) (to H.V.R.F) and financed in part by the Coordenação de Aperfeiçoamento de Pessoal de Nível Superior - Brasil (CAPES) – Finance Code 001 [Grant Number 88887.928702/2023-00] (to J.V.S. G) and the National Institutes of Health Grant GM144083 (to B.G.P). The opinions, hypotheses, and conclusions or recommendations expressed in this material are the responsibility of the author and do not necessarily reflect the views of FAPESP. The funders had no role in study design, data collection and analysis, decision to publish, or preparation of the manuscript.

**Competing interests:** The authors have declared that no competing interests exist.

against cancer. Leveraging breakthroughs in deep learning within the biosciences, this study explores whether deep learning architectures for general protein design, such as ProteinMPNN and ESM-IF1, can design TCRs to bind target antigenic peptides presented by the MHC (pMHC) based on fixed-backbone structures of the complexes. We utilized a comprehensive set of sequence-based and structure-based metrics, derived from deep learning or classical physics-based principles, to evaluate the generated TCR designs. Our findings highlight the benefits of these methods compared to traditional physics-based design approaches and pinpoint deficiencies for improvement. The results contribute to advancements in the field of TCR design, serving as a guiding framework for the structure-based design of TCRs using deep learning, thereby providing the basis for computational engineering of specific TCRs for T cell-based immunotherapy.

## Introduction

T cell receptors (TCRs) are heterodimeric receptors located on the T cell surface, specialized in recognizing and binding to antigenic peptides presented by the major histocompatibility complex Class I (MHC-I) or Class II (MHC-II). The peptide recognition by the TCR enables T cells to identify foreign proteins from infectious organisms, as well as altered proteins in tumor cells, and respond accordingly. The canonical binding of TCR to peptides presented by the MHC (pMHC) canonically involves the TCR complementary-determining region (CDR) loops (CDR1, CDR2, and CDR3) of both heterodimeric chains (α and β chains, for instance). The key specificity in TCR binding primarily residues in the interaction of CDR3 loops, particularly the CDR3β, which is the most variable region of the TCR as a consequence of random rearrangements of variable (V), diversity (D), and joining (J) gene segments [1].

Given the pivotal role of T cells in orchestrating the adaptive immune response, substantial efforts have been employed to genetically engineer T cells capable of expressing TCRs designed to recognize specific antigens. This strategy, known as TCR-engineered T cell (TCR-T) therapy, has emerged as a notable route for immunotherapy, particularly in the context of cancer [2–4]. When compared to other immune response-based therapies like antibody and chimeric antigen receptor T cell (CAR-T) therapies, TCR-based approaches offer distinct advantages centered on their capacity to efficiently recognize a wide universe of antigens from various subcellular compartments, extending beyond surface-expressed targets [5]. Over the past few years, no fewer than 25 phase I/II clinical trials involving TCR-T cell therapy, targeting diverse epitopes, have been reported for the treatment of solid tumors, emphasizing the significant efforts in this field [4]. In 2022, the FDA approved tebentafusp, the first TCR-based therapy to target the gp100 epitope, used to treat HLA-A*02:01-positive adult patients with unresectable metastatic uveal melanoma [6]. Promising results have also been observed in clinical trials for high-affinity engineered TCRs such as afamitresgene autoleucel, which targets the MAGE-A4 melanoma-associated antigen presented by HLA-A*02 [7].

Despite the advancements in TCR-T therapy, the intricate and still poorly understood mechanism of antigen recognition by TCRs and subsequent T cell activation present challenges to the rational design of TCRs for therapeutic applications. Unlike antibodies, which undergo an affinity maturation process involving somatic hypermutations to enhance affinity for the target, natural TCRs typically exhibit low affinities in the micromolar range [8]. In certain scenarios, such as therapies targeting tumor-associated antigens, the practical application of natural TCRs is limited due to the elimination of T cells recognizing self-antigens during thymic selection [4, 9]. To address this limitation, enhanced affinity TCRs can be generated

using experimental techniques, such as phage display [10–13]. However, extreme increases in TCR affinity to supraphysiological levels are associated with impaired T cell activation [14]. Furthermore, substantial increases in TCR affinity for a specific target may lead to increased affinity for other unintended targets, resulting in cross-reactivity. For instance, cross-reactivity with self-antigens can result in off-target effects, as observed in a notable case where the use of a TCR with improved affinity for binding the MAGE-A3 antigen from cancer cells induced cardiac toxicity in treated patients due to the high similarity of this peptide to a self-peptide from the cardiac Titin protein [15, 16]. Given the potential drawbacks associated with supraphysiological affinities, the development of TCRs with low micromolar affinities is advocated as a viable solution.

In this context, solved 3D atomic structures or molecular models of TCRs in complex with antigenic peptides bound to the MHC provide valuable details about the TCR recognition mechanism and can serve as the basis for the rational design of TCRs through computational tools. By leveraging these 3D structures, a refined balance between affinity and specificity can be achieved by focusing the TCR design toward peptide contact regions rather than the MHC, thereby mitigating cross-reactivity [17]. Structure-based computational design approaches utilizing physics-based scoring functions have been successfully employed in enhancing the affinity of the DMF5 TCR against melanoma-associated antigens, yielding a remarkable 400-fold increase in binding affinity [18]. However, it is noteworthy that these scoring functions face challenges in generalizing across various TCR:pMHC complexes [19], necessitating calibrations to enhance accuracy despite their significance in TCR optimization.

On the other hand, the growing availability of general protein 3D structures and sequences, alongside recent advancements in machine and deep learning techniques, has significantly accelerated the accurate prediction of protein amino acid sequences capable of folding into a specific backbone structure. Within this context, ProteinMPNN is revolutionizing the protein design field by leveraging the power of Graph Neural Networks and Message-Passing algorithms, alongside an extensive training set of protein structures, demonstrating remarkable capabilities in designing entirely novel protein sequences [20]. Additionally, the ESM inverse fold (ESM-IF1) model, trained on 12 million AlphaFold2-modeled protein structures combined with experimentally determined ones, using Graph Neural Networks with Geometric Vector Perceptron layers, has shown promising results in predicting protein sequences that fold into a predetermined backbone structure [21, 22]. While the applicability of these models has been demonstrated in the design pipeline of protein binder interfaces [20, 21, 23], their potential in TCR design remains unexplored.

In this study, we extensively explored the use of ProteinMPNN and ESM-IF1 for structure-based TCR interface design, encompassing various design scenarios and comparing them with a widely used physics-based method, Rosetta Design [24]. To assess the efficacy of these deep learning models, we compared the designs with known TCRs by basing the design process on experimentally solved TCR structures in complex with the pMHC. We employed a comprehensive and orthogonal set of evaluation metrics, beginning with sequence-based metrics that included the percentage of native sequence recovery and physicochemical similarity at interface positions. Recovery of native sequence is a widely used evaluation metric allowing to compare TCR design recoveries with the recovery of general proteins designed in other studies [20, 21, 25–27] and valuable in the TCR interface design context for understanding whether the methods recapitulate natural sequence features of TCRs that determine CDR loop folding and the general low binding affinity. Using sequence information, we also estimated the probability of the CDR3s designed by ProteinMPNN and ESM-IF1 being generated by a natural V(D)J recombination process using the Optimized Likelihood estimate of immunoGlobulin Amino-acid sequences (OLGA) algorithm [28]. However, recognizing the limitations of the sequence

recovery metric in evaluating TCR designs and the importance of sequence diversity in the design process, we leveraged the diversity of generated sequences and employed a comprehensive set of structure-based evaluation metrics that are independent of similarity to native sequences. Initially, we modeled the design sequences with TCRModel2 [29] and assessed the confidence of the models. The TCR designs were further evaluated using Rosetta physics-based metrics to assess interface quality and complex stability. Finally, to compare the binding affinity of the designed complexes with native TCRs, we employed robust protocols of Molecular Dynamics (MD) simulations coupled with Molecular Mechanics Poisson-Boltzmann Surface Area (MM/PBSA) free energy calculations. To ensure accurate binding affinity prediction, this protocol underwent benchmarking against a curated dataset of TCR:pMHC mutants with experimental binding affinity measurements.

We found that ProteinMPNN and, particularly, ESM-IF1 demonstrated a remarkable ability to recover native TCR interface amino acid sequences compared to a classical physics-based method, successfully reproducing the amino acid composition of this interface. Importantly, this recovery was not solely based on the backbone or intra-TCR contacts but heavily relied on specific interactions with the pMHC. TCR designs exhibited modeling and energetic scores comparable to native structures. Despite the majority of designs showing similar predicted binding affinities, we identified some designs with improved affinity compared to the native TCR.

We aim for this study to serve as a guiding framework for the structure-based design of TCRs using deep learning and thus provide the basis for the computational engineering of specific TCRs targeting antigenic peptides presented by the MHC, such as tumor and viral antigens.

## Materials and methods

### Benchmarking dataset

To assess the capability of computational design methods for TCR designing, we curated a dataset of TCR:pMHC complexes, involving both MHC-1 and MHC-II, with solved 3D structures that are not included in the ProteinMPNN training dataset (August 31, 2021, date cut-off) and not in AlphaFold2.3 (September 30, 2021, date cut-off). This date cut-off ensures that the complexes under consideration were not part of ProteinMPNN or AlphaFold2.3 training datasets, which could potentially lead to an overestimation of sequence recovery or modeling accuracy results. ESM-IF1 training involves the utilization of protein models derived from UniRef sequences and experimentally determined structures from the CATH 4.3 database. However, it's important to note that the CDR3 sequences, which are central to this study, are not included in the UniRef dataset as they are generated through gene rearrangement. Additionally, the CATH 4.3 release occurred in 2019, which falls within the data cut-off period for ProteinMPNN and AlphaFold2.3.

The structures were collected from the TCR3d database [30] on December 13, 2023, and selected based on a resolution cut-off of 3.25 Å. We verified whether the chosen TCR:pMHC complexes had a related PDB entry released before the cut-off dates. For this reason, the CDR3β sequences of the TCR from each complex were submitted to a blast search using the blastp server [31]. Additionally, we verified redundancies within the selected dataset of TCR:pMHC complexes (considering the same TCR, peptide and MHC). For this reason, complexes with different peptide or MHC sequences, despite having the same CDR3, were maintained in the dataset (S1 Table). Structures with unnatural amino acids were excluded from the dataset. Following these filtering steps, a total of 32 and 6 TCR:pMHC structures (MHC-I and MHC-II, respectively) were selected, as listed in S1 Table.

## ProteinMPNN and ESM-IF1 design protocols

For ProteinMPNN design, we used the v_48_020 model with default backbone noise (0.00) and sampling temperature of 0.1 to generate 10 designs per each test case, unless otherwise specified. The ProteinMPNN code used for running the desing was obtained from https://github.com/dauparas/ProteinMPNN. To evaluate the effects of sampling temperature on sequence generation, we utilized a range of temperatures from T = 0.000001 to 5. Additionally, to assess the impact of the number of generated sequences, we varied the quantity of generated sequences per test case from 5 to 500. A total of five design strategies were evaluated: (1) restricted the design to CDR3 (α and β TCR chains) within a proximity of 5 Å to either the peptide or MHC, (2) CDR1, CDR2, or CDR3 positions (α and β TCR chains) within a 5 Å distance to the peptide or MHC, or (3) the entire CDR3 (α and β TCR chains), (4) all CDR1, CDR2 and CDR3 positions (α and β TCR chains) and (5) all TCR variable domain positions (α and β TCR chains). Positions outside the specified criteria were held constant, and for the designed positions, all amino acid substitutions, including cysteine, were allowed. All tests were performed maintaining the same seed.

For ESM-IF1 design, we employed the esm_if1_gvp4_t16_142M_UR50 (fair-esm v2.0.1) model with sampling temperature of 0.2 to generate 10 designs per each test case, unless otherwise specified. Similarly to ProteinMPNN, effects of sampling temperature on sequence generation were evaluated using a range of temperatures from T = 0.000001 to 5, and the quantity of generated sequences per test case varied from 5 to 500 to assess the impact of the number of generated sequences. Five design strategies were explored: (1) restricting the design to CDR3 (α and β TCR chains) within a proximity of 5 Å to either the peptide or MHC, (2) including CDR1, CDR2, or CDR3 positions (α and β TCR chains) within a 5 Å distance to the peptide or MHC, (3) encompassing the entire CDR3 (α and β TCR chains), (4) all CDR1, CDR2 and CDR3 positions (α and β TCR chains) and (5) all TCR variable domain positions (α and β TCR chains). Positions outside the specified criteria were held constant. The designed positions were changed to `<mask>` tokens, and all amino acid substitutions, including cysteine, were allowed. Given the multi-chain structure, a padding of 10 `<pad>` tokens was used to separate the chains. The ESM-IF1 design protocol used in our study is available at https://github.com/LBC-LNBio/ESMIFDesign. The original ESM-IF1 code is available at https://github.com/facebookresearch/esm.

In the design process CDR1, CDR2 and CDR3 positions were delimited based on Aho numbering scheme: CDR1 (27–40), CDR2 (58–70) and CDR3 (106–139). In both methods, ProteinMPNN and ESM-IF1, we utilized the default decoding processes. For ESM-IF1, sequences are sampled autoregressively. In contrast, ProteinMPNN employs an order-agnostic autoregressive approach where the decoding order is randomly determined. Except for the tests where we removed the pMHC, in all design strategies we used the pMHC context (the TCR bound complex) for designing the TCR.

## Rosetta design protocol

For Rosetta design, we utilized Rosetta version 3.5.1, and the protocol was executed using RosettaScripts. The target design positions were the same as those in deep learning-based methods, and we utilized a Rosetta resfile to indicate the positions and allowed substitutions (in this case we excluded cysteine) (resfile example in S1 Appendix). Throughout the mutation process, we applied the InterfaceDesign2019 fast relax protocol. A total of 10 designs were generated per test case. Alternatively, we generated a total of 1000 designs per test case and ranked the designs based on the Interface Analyzer *dG_separated* term or the *total_score* term,

considering the ref2015 score function to identify the top 10 best-scored designs. A representative example of the RosettaScripts xml file is provided in S2 Appendix.

## Design quality assessment at sequence level

To evaluate the success of designs, we compared the designed sequence to the native sequence by using the sequence recovery metric. The sequence recovery (Eq 1) measures the percentage of positions in designed sequences that match the amino acids in the native sequence. Given two sets of corresponding positions $X = \{x_1, x_2, \ldots, x_n\}$ and $Y = \{y_1, y_2, \ldots, y_n\}$ of equal length $n$, the sequence recovery is defined as:

$$\text{Sequency recovery } (\%) : (X, Y) \mapsto 100 \times \frac{1}{n} \sum_{i=1}^{n} \delta(x_i, y_i) \in [0, 100] \tag{1}$$

where $\delta(x_i, y_i)$ is defined as:

$$\delta: (x_i, y_i) \mapsto \begin{cases} 1 & \text{if } x_i = y_i \\ 0 & \text{if } x_i \neq y_i \end{cases} \tag{2}$$

The percentage sequence recovery of 100% indicates that the designed sequence is identical to the native sequence, and 0% indicates no recovery. When evaluating the sequence recovery, we only considered non-redundant sequences with each test case. The percentage of non-redundant sequences generated by ProteinMPNN and ESM-IF1 is represented by the Uniqueness metric (described below), as shown in S5 Fig, considering the corresponding sampling temperature.

When comparing ProteinMPNN and ESM-IF1 across different sampling temperatures, we employed two additional sequence metrics: Uniqueness and Entropy. The Uniqueness (Eq 3) represents the percentage of unique sequences within the designed set. Given a set of $m$ designed sequences $\mathcal{D} = \{D_1, D_2, \ldots, D_m\}$, with $u$ denoting the count of unique sequences, the Uniqueness is calculated as:

$$\text{Uniqueness } (\%) : (u, m) \mapsto 100 \times \frac{u}{m} \in \left[\frac{100}{m}, 100\right] \tag{3}$$

The Entropy relies on Shannon's information-theoretic entropy (Eq 4) and measures the diversity of the designed sequences. For each position $j$ in the designed sequences of length $L$, the probability distribution of amino acids is estimated from the observed frequency. The Shannon's information-theoretic entropy of the position $j$ in the designed sequences is defined as:

$$H_j: (\mathcal{D}) \mapsto - \sum_{i=1}^{20} p_i \log_2(p_i) \in [0, \log_2(20)] \tag{4}$$

where $p_i$ is the probability of amino acid $i$ at position $j$ in the designed sequences.

The entropy of each position is then averaged across all positions to obtain the Mean Sequence Entropy (MSE) as defined in Eq 5:

$$MSE: (\mathcal{D}) \mapsto \frac{1}{L} \sum_{j=1}^{L} H_j \in [0, \log_2(20)] \tag{5}$$

Additionally, we explored physicochemical similarity by considering changes from an amino acid in the native sequence to another in the design sequence within the same group.

Amino acid groups were created using hierarchical clustering of BLOSUM62 amino acid features into 6 groups ([A, G, S], [C], [D, E, P, T], [Q, N, H, R, K], [I, L, M, V], [F, Y, W]).

## Assessment of naturalness of V(D)J recombination

To evaluate the probability of designed CDR3 sequences being generated by the natural V(D)J recombination process, we utilized the Optimized Likelihood estimate of immunoGlobulin Amino-acid sequences (OLGA) [28]. We installed OLGA from https://github.com/statbiophys/OLGA and executed it via the command line using the flags humanTRA or humanTRB to compute the generation probabilities for CDR3α and CDR3β, respectively, with all other parameters set to default.

## Assessment of TCR sequence variability and TCR interface contacts

To understand the amino acid variability across the TCR variable domain sequence, we compiled a set of human TRAV, TRAJ, TRBV, TRBJ, and CDR3 sequences. We collected 113 TRAV domain alleles, 147 TRBV domain alleles, 71 TRAJ, and 16 TRBJ alleles from IMGT, including all functional, open reading frame, and pseudogene alleles (https://www.imgt.org/vquest/refseqh.html). The sequences were renumbered according to the Aho numbering scheme using ANARCI [32] for alignment in sequence logos and CDR annotations. TRAJ and TRBJ sequences underwent multiple sequence alignment (gap opening of 10 and gap extension of 4) using the R *msa* package. We collected 19,948 human CDR3α and 20,662 CDR3β sequences with known antigen specificity from VDJdb [33] in November 2023. To number the CDR3s according to Aho, we first constructed each full TCRα or TCRβ sequence using Stitchr [34], then applied ANARCI to number the complete sequence. Subsequently, only the CDR3 positions, including gaps, were extracted. The sequence logos were generated in R using the *ggseqlogo* package [35].

To compute the contacts made by the TCR variable domain with the peptide or MHC in the MHC-I and MHC-II test cases, we utilized the *Biopython* package [36]. A contact was defined based on a distance threshold of 5 Å involving all atoms.

## Identification of TCR buried and hotspot positions

To compute sequence recovery over buried positions in the TCR:pMHC complexes of the test cases, we identified buried positions in native structures based on the relative solvent accessibility (RSA). Initially, the solvent-accessible surface area for each position was determined using the *FreeSASA* Python package. Then, the ratio between the calculated accessible area and the maximum allowed accessibility for each amino acid was estimated to obtain the RSA. We classified a position as buried if its RSA was lower than 0.2. The reference maximum accessibilities were obtained from [37].

The hotspot positions were determined by performing an alanine scanning calculation using Rosetta 2.3 software. In this analysis, a residue was considered a hotspot if the absolute calculated ΔΔG for alanine substitution was greater than 0.5 kcal/mol. The alanine scanning command is provided in S3 Appendix, along with an example of the mutation list file used in this protocol (S4 Appendix).

## Design quality at structural level

To evaluate the structural quality of ProteinMPNN or ESM-IF1 TCR designs, we modeled these designs and employed various orthogonal approaches for assessment. This included AlphaFold2-based modeling, Rosetta-based scores and molecular dynamics simulations with

MM/PBSA calculations. Further details regarding the latter are provided in the following section.

The ProteinMPNN or ESM-IF1 design sequences were modeled with TCRModel2, a modified version of AlphaFold2 focused on TCR modeling [29]. The standalone version of TCRModel2, available at https://github.com/piercelab/tcrmodel2/, was employed with default parameters, except for the *max_template_date* flag, which was set to 2021-09-30 (AlphaFold2.3 cut-off date). Relaxation with Amber was allowed. The confidence score, which is a weighted combination of ipTM and pTM (ipTM*0.8 + pTM*0.2) [38], was extracted from the TCRModel2 output statistics file. The RMSD between the CDR3 or all CDRs of the design structure and the native structure was estimated considering the CDR3 or all CDRs backbone atoms, respectively, after superposition of the design structure to the native structure by the MHC. RMSD calculations were performed in R using the *Bio3D* package [39] and the DockQ calculation was performed using its standalone version obtained from https://github.com/bjornwallner/DockQ [40].

For the assessment of the TCR:pMHC interface and complex quality, we employed Rosetta Interface Analyzer, utilizing Rosetta software (version 3.5.1), with the entire protocol executed through RosettaScripts. Upon mutations of the native structures according to the designed amino acids, the complex was relaxed with the FastRelax protocol using backbone constraints to prevent structure displacement. The REF2015 score function was applied throughout the protocol, with *dG_separated* and *total_score* terms extracted from the output file. An example of the RosettaScripts used for this purpose is provided in S5 Appendix.

Random dissimilar TCR sequences were generated by substituting the native amino acids at the designed positions with randomly selected amino acids from different groups. Amino acid groups were defined by clustering the amino acids into six groups based on hierarchical clustering of BLOSUM62 features, the same used for computing sequence similarity.

## TCR:pMHC binding affinity dataset and benchmark

To validate the capability of the binding affinity methods employed in this study to accurately predict TCR:pMHC interaction and stability, we created two different datasets of TCR:pMHC complexes. The first set comprised solved 3D structures of wild-type and mutant pairs, along with their corresponding binding affinity data. The second set used the same wild-type solved structures but expanded the number of mutants by modeling them, rather than using solved structures.

To construct the first dataset of TCR:pMHC structures, we selected wild-type and mutant pairs from the ATLAS server [41] and excluded from the dataset samples in which the experimental binding affinity of the wild-type and mutant complex were not obtained from the same study. By doing this, we seek to minimize noise included from different methods or experimental setups. Following these criteria, 12 TCR:pMHC complexes were found, forming 7 wild-type and mutant pairs (S2 Table). Since native and mutant pairs could have different sequence lengths as a consequence of N or C terminal electron density confidence, we trimmed the longest sequence in the pair to match the length of the shorter sequence. This was done in order to avoid bias created by the differing number of residues in MD simulations and MM/PBSA calculations.

To construct the second dataset, we selected mutants from the same wild-type complexes used in the first dataset, sourced from ATLAS, but lacking solved 3D structures. We only included mutants whose binding affinities were determined in the same study as their corresponding wild-types. To ensure a broad range of binding affinities for analysis, we excluded mutants with redundant binding affinities, applying a cutoff difference of 0.1 kcal/mol. This

process resulted in a total of 35 mutants across 5 different wild-type complexes, as detailed in S3 Table. Mutations were introduced using *PyMOL* package following the same procedures as for TCR designs, with specifics provided in the subsequent section.

## Molecular dynamics simulations

To estimate the binding affinity of the TCR:pMHC complexes, we first sampled the conformational space of the complexes with molecular dynamics (MD) simulations.

For MD simulations of native and designed TCRs in complex with the pMHC, we obtained the native TCR:pMHC complexes from the Protein Data Bank (PDB). The complexes were composed by the variable and constant TCR domains, including both α and β chains, the peptide, the MHC and the β2-microglobulin. None of the selected complexes had gaps in the structure. The native complex was processed by removing water and small molecules. Protonation state of residues was determined at a pH of 7.0 using Propka 3.0 [42] with pdb2pqr [43]. The designed TCR complexes were built using the native structures as reference and substituting the designed residues with *PyMOL* Python package. The protonation states of histidine at non-designed positions in designed complexes were maintained the same as the one in the corresponding native complex.

Prior to protein solvation, water molecules were placed on water sites by following a protocol similar to [44]. Specifically, the three-dimensional reference interaction site model (3D-RISM) [45, 46] and the MoFT suite [47] were used to calculate the solvent density around the protein and to find and place discrete water molecules into the most favorable locations, respectively. This was followed by aligning the complexes with its principal axis of inertia to the z axis, in order to reduce the number of waters using cpptraj from AmberTools23 [48]. All structures were solvated in a rectangular water box such that water molecules encompass the box boundary within 14 Å of the protein and were subsequently neutralized by the addition of $Na^+$ ions. The Amber ff14SB force field [49] and the TIP3P water model [50] were used for protein and water, respectively. Disulfide bonds were manually bonded using the information derived from the pdb4amber module from AmberTools23 [48]. The topology and initial coordinates for each solvated system were done using TLeap from AmberTools23 [48].

MD simulations were performed using a graphics processing unit (GPU) accelerated Amber from AMBER22 [51, 52]. Each system was subjected to minimization, equilibration, and production following the protocol reported in [44]. Prior to replicas production, a minimization followed by a short MD simulation was conducted. This involved heating the system from 50 K to 298 K and restraining the protein atom positions. Subsequently followed, the entire system is minimized without any restrictions. The process of generating replicas per complex involved seven steps: (1) the system was heated from 25 K to 298 K for 20 ps at NVT with restraints on the CA atoms with a force constant of 5 kcal/mol/$Å^2$, (2–5) four NPT simulations of 10 ps, restraining the CA atoms with a decreasing force constant from 4 kcal/mol/$Å^2$ to 1 kcal/mol/$Å^2$ (6) a 1 ns NPT without any restraint, and (7) a 3 ns NPT production run at 298 K. A total of 15 replicas were run for each system. The production simulations were used for MM/PBSA calculations, as well as PCA and RMSD analysis. The temperature was controlled with a Langevin thermostat [53] using a collision frequency of 1 $ps^{-1}$, while the pressure was set to 1 bar using a Berendsen thermostat [54] with a relaxation time of 1 ps.

The MD simulations of the benchmarked of solved complexes, described in S2 Table, followed the same protocol used above for the native complexes. In this case, no mutation needed to be performed since both native and mutant TCR:pMHC complexes were available. For benchmarking the modeled mutants, we employed the same strategy detailed above for the TCR designs.

MD simulations of both designed and native TCRs in complex with pMHC were analyzed by computing the root-mean-square deviation (RMSD) of the CDR3 backbone atoms ($\alpha$ and $\beta$ chains, considering only a range of positions from the first designed positions to the last designed positions) between trajectory frames from production runs and the corresponding native crystal structure. This involved superposing the trajectory frames by the MHC of the corresponding native crystal structure. Both superposition and RMSD calculations were conducted using the R *Bio3D* package [39] in RStudio.

To assess the conformational space sampled by the designs compared to the native TCRs, we employed principal component analysis (PCA). The PCA was carried out using the pca.xyz function of the R *Bio3D* package [39] in RStudio, utilizing the coordinates of the CDR3 positions from the superposed trajectory used in RMSD calculations. All systems, their respective replicas and designs were included in the calculation. The first two principal components were utilized in the evaluation of conformational space sampling.

## Free energy calculations with MM/PBSA

The MM/PBSA calculations were performed over the 15 MD simulations replicas of 3 ns saving every 10 ps, giving a total of 300 frames each. For each system, a total of 4500 frames were used for the calculation. The ionic strength and the internal dielectric constant were set to 0.15 M and 6, respectively. All other options were set to default. The MPI version of the MMPBSA. py [55] from AmberTools23 [48] was used for free energy calculations through MM/PBSA approximation. Unlike [44], no explicit waters at the interface nor entropy corrections were included in the MM/PBSA calculations.

## Results

### Deep learning-based methods outperforms physics-based methods in recovering TCR native sequences

To assess the capability of deep learning and physical-based methods in designing TCR sequences, based on 3D structures, for targeting specific pMHC complexes, we curated a structural dataset consisting of non-redundant TCR:pMHC complexes (32 MHC-I complexes and 6 MHC-II complexes; see Methods section for details). These complexes served as native reference structures for comparison with the generated designs. Descriptive analyses of both set of structures, including the solvent accessibility across the TCR chains in the bound state and the TCR contacts with the peptide and MHC, are presented in S1 and S2 Figs, respectively, and can provide insightful views of these complexes and interfaces in the design context. Most TCR contacts with the peptide originate from the CDR3, especially the CDR3$\beta$, which guides the specificity of this interaction (S1 Fig). Due to the canonical binding mode of the TCR, the CDR3 is typically buried at the pMHC interface, except for longer CDR3 loops that expose the central part to solvent, followed by CDR1 and CDR2 (S2 Fig). To provide context for the TCR sequence variability in our TCR design analysis, we included a per position variability analysis of the TCR variable domain, encompassing CDR3 sequences, in S3 Fig.

As a first TCR design scenario, we focused on designing the TCR CDR3 positions ($\alpha$ and $\beta$ chains) at the interface with the pMHC (Fig 1A). During the design process, the pMHC and TCR non-designed residues were provided as context for the methods analyzed. Considering this scenario, on average, we designed 10 and 11 positions of the CDR3s per each TCR: pMHC-I and MHC-II test case, respectively (S4 Fig).

Our initial assessment of design success involved comparing the identity of amino acids generated at the designed positions to the native amino acids, computing the percentage of

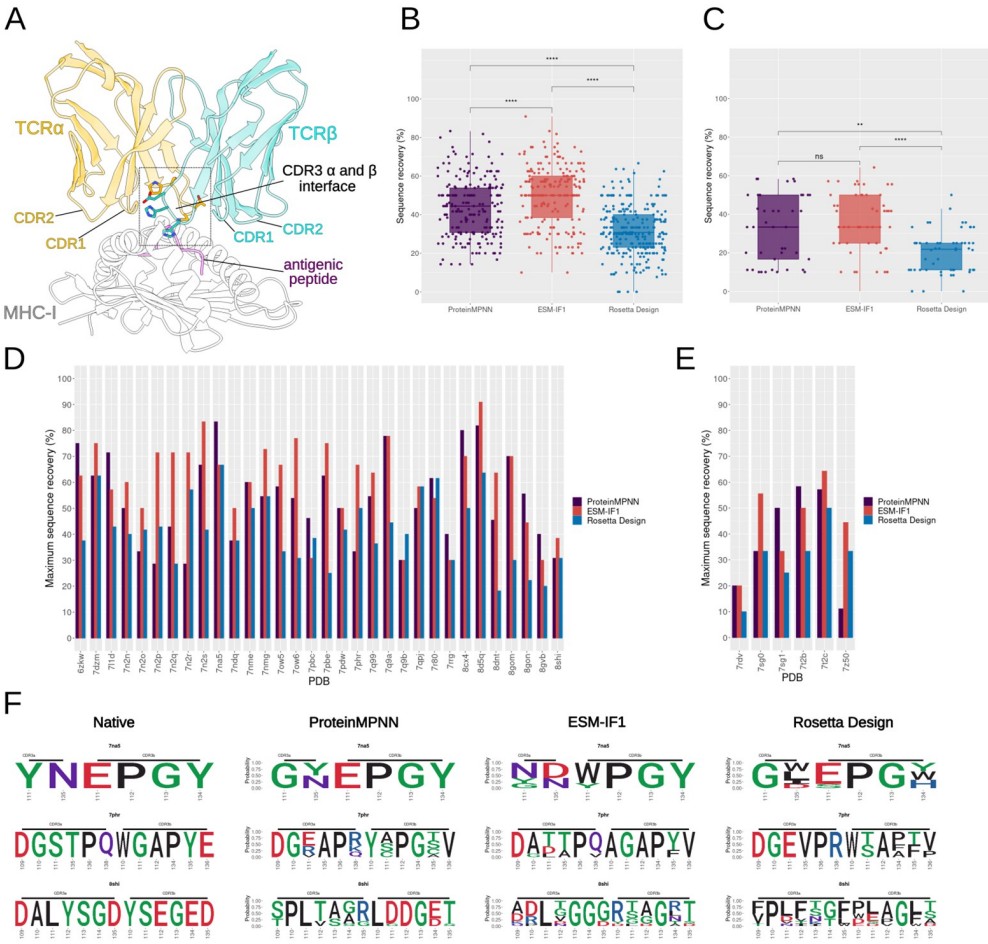

**Fig 1. Sequence recovery analysis of interface CDR3s amino acids in designs with ProteinMPNN, ESM-IF1 or Rosetta Design (InterfaceDesign2019 protocol). (A)** Representative structure of a TCR:pMHC complex (PDB ID: 7nme). The TCR variable and MHC chains were trimmed to just include components spatially related to the interface. The interface is indicated and the CDR3s amino acids composing the interface are shown as sticks. **(B)** Percentage of sequence recovery per method considering all MHC-I test cases. Each point represents a unique design sequence from a test case. For each test case, a total of 10 designs were generated by each method, but redundant designs were removed from the plot. For ProteinMPNN we employed a temperature sampling of 0.1, whereas for ESM-IF1 a temperature sampling of 0.2 was used (see Methods). Statistical two-sample pairwise comparison between methods were performed using Mann-Whitney test with the R *ggpubr* package. Significance is indicated above each box plot (**** and ** correspond to a p-value below 0.0001 and 0.01, respectively, while 'ns' means no significance). **(C)** same as (B), but for MHC-II. **(D)** Maximum sequence recovery obtained for each MHC-I test case and **(E)** for MHC-II test cases. **(F)** Sequence logo of three MHC-I test cases: 7na5, 7qhr, and 8shi. Each row of the panel corresponds to a specific test case and each column corresponds to the design method applied. The first column presents the native amino acids.

sequence recovery for each test case. A critical parameter that affects sequence design, especially the diversity of generated sequences, in deep learning-based methods is temperature sampling. For ProteinMPNN, we employed a temperature of 0.1, which was used by its developers and others [20, 23]. In our assessments, this temperature achieved the highest sequence recovery while producing higher sequence entropy and uniqueness than lower temperatures (S5 Fig). To ensure comparability, we chose a temperature (T = 0.2) for ESM-IF1 that resulted in sequences with similar sequence entropy to those from ProteinMPNN at T = 0.1. These temperatures for ProteinMPNN and ESM-IF1 were also consistent in terms of the percentage of uniqueness (S5 Fig).

On average, ProteinMPNN and ESM-IF1 achieved sequence recovery percentages of 43.9% and 50.1%, respectively, for MHC-I, and 32.0% and 35.6% for MHC-II cases, with each test case generating 10 designs (Fig 1B and 1C). Sequence recovery percentages per target are presented in S4 Fig. Notably, both deep learning-based methods, ProteinMPNN and ESM-IF1, surpassed the sequence recovery of designs generated by Rosetta FastDesign protocol [24], with sequence recovery percentages of 31.7% and 19.6% for MHC-I and MHC-II cases, respectively. In 66% and 84% of test cases (including both MHC-I and MHC-II), ProteinMPNN and ESM-IF1, respectively, generated designs with superior sequence recovery compared to Rosetta Design. As an alternative to the default Rosetta FastDesign approach, we also explored protocol variations. This included generating 1000 designs and calculating the sequence recovery over the top 10 designs ranked by Rosetta energy terms. However, we observed consistent trends across these variations (S6 Fig).

When comparing ProteinMPNN and ESM-IF1, the latter generated designs with higher sequence recovery than ProteinMPNN in 55% of cases, while ProteinMPNN generated higher sequence recovery rates in 29% of cases. On average, ProteinMPNN, ESM-IF1, and Rosetta FastDesign achieved maximum sequence recovery of 53.6%, 60.6%, and 41.5%, respectively, for MHC-I, and 38.3%, 44.6%, and 30.8% for MHC-II (Fig 1D and 1E). A Pearson correlation of 0.51 was observed when comparing the maximum sequence recovery of MHC-I test cases obtained by ProteinMPNN and ESM-IF1 (S7 Fig). Most outliers arose from ESM-IF1 showing improved sequence recovery in comparison to ProteinMPNN. However, there were also test cases where higher maximum sequence recovery was achieved by ProteinMPNN. Together with the fact that no single designed sequence was found in common between ProteinMPNN and ESM-IF1 for the corresponding test cases, these findings suggest that combining designs from both methods may offer advantages, particularly in terms of sequence recovery and increasing diversity.

We explored sequence and structural features that might explain the superior performance of either ProteinMPNN or ESM-IF1 in certain test cases. The test cases were divided into two groups: one where ESM-IF1 achieved higher maximum recovery (13 cases) with a significant improvement of at least 10%, and another where ProteinMPNN did so (8 cases). Analyzing the per-position frequency of native amino acids in the CDR3s from these test cases, we observed a slight increase in the preference of certain amino acids such as at positions 110 and 111 of CDR3β from ESM-IF1 group, but no clear pattern emerged that could explain the differences in maximum sequence recovery (S7C Fig). Additionally, the length of the CDR3s in both groups was not significantly different (S7D Fig), nor were the buried surface area and shape complementarity of the TCR interaction interface (S7E Fig).

When analyzing the composition of native amino acids at positions exclusively recovered by ProteinMPNN, we found a bias toward serine and threonine, suggesting these amino acids are more readily recovered by ProteinMPNN (S7F Fig). Conversely, aspartic acid and histidine were more easily recovered by ESM-IF1 compared to ProteinMPNN. However, this may not fully account for the observed differences in maximum recovery. For instance, in the 7n2p case, which showed the largest difference (71% recovery by ESM-IF1 versus 28% by ProteinMPNN), only about half of the native amino acids that presented preferences for ESM-IF1 recovery (phenylalanine, alanine, and aspartic acid) were present in the 7n2p designed positions (native non-contiguous amino acid sequence FATYSTD encompassing CDR3α and CDR3β). Furthermore, the balance between recovery from CDR3α and CDR3β was maintained in both ProteinMPNN and ESM-IF1. For positions recovered exclusively by ProteinMPNN, 40% were from CDR3α and 60% were from CDR3β. These percentages were consistent for positions recovered exclusively by ESM-IF1, indicating no bias toward specific TCR chains that would explain the recovery differences.

Moreover, no substantial difference was observed in the composition of amino acids in the local environment of positions recovered exclusively by ProteinMPNN or ESM-IF1 (S7G Fig). While the difference in maximum sequence recovery may result from factors not analyzed here, it is possible that the observed differences arise from the sequences or structures encountered during model training.

Given the limited number of MHC-II test cases, we tested if the observed reduced sequence recovery of MHC-II test cases might simply reflect a statistical sampling bias. We computed the cumulative probabilities of the MHC-II maximum sequence recovery averages for ProteinMPNN, ESM-IF1, and Rosetta Design under the distributions of MHC-I maximum sequence recovery averages, constructed by sampling averages from 6 random test cases (the same number as MHC-II test cases) 1000 times. We obtained two-tailed p-values of 0.012, 0.008, and 0.023 for ProteinMPNN, ESM-IF1, and Rosetta Design, respectively. These results indicate that the maximum averages observed for MHC-II test cases significantly differ from those observed for MHC-I. It is not clear whether the difference in MHC-I and MHC-II recovery percentage is related to particular structural features or underrepresentation in training dataset. Future analysis with additional MHC-II test cases will likely shed light on the reasons behind the observed differences in sequence recovery. Given the more comprehensive number of available 3D structures, we focused the subsequent analysis on only the MHC-I test cases. Designed amino acids of three MHC-I test cases from ProteinMPNN, ESM-IF1, and Rosetta are represented as sequence logos in Fig 1F.

Increasing the number of generated designs (up to 500) increased the maximum sequence recovery for some MHC-I test cases (S8 Fig), reaching an average maximum sequence recovery of 59.9% (6% higher than 53.6% with 10 sequences sampled, but not significantly) and 69.1% (9% higher than 60.3% with 10 sequences sampled) for ProteinMPNN and ESM-IF1, respectively. Although this increase in sequence recovery was not consistently observed across all test cases (S8 Fig), these findings suggest that, at least for some cases, expanded sampling may contribute to the generation of designs closer to the native sequence.

From a practical standpoint, in silico generated designs usually need to be ranked and prioritized for subsequent robust in silico evaluation or experimental validation. Particularly for ProteinMPNN, which provides a score for each design sequence, we tested to rank their designs in the expanded sampling set by the provided ProteinMPNN score (defined as a negative average log-probability) to obtain the top 10 best-scored designs. Although in some cases, such as 8d5q, we observed designs with lower scores (the lower the score, the higher the confidence on the design prediction), achieving high sequence recovery rates, this trend was not consistently observed across all test cases (S9 Fig). Additionally, when correlating the scores of all designs with the sequence recovery, we obtained only a moderate correlation (-0.44; S9B Fig). When comparing the sequence recovery of the top 10 ranked designs with the previous 10 designs generated by ProteinMPNN, we did not observe significant differences (average of 44.9% in comparison to the default protocol's 43.9%) (S10 Fig). Together, these findings indicate that, at least for the CDR3 interface design strategy, 10 designs generated by ProteinMPNN would be sufficient to achieve suitable sequence recovery rates.

To investigate the contribution of each designed position to the sequence recovery, we computed the sequence recovery per CDR3 position (Fig 2A). In general, the sequence recovery was uniformly distributed across the CDR3 positions. In CDR3α, position 133, typically located in the middle of the CDR3 loop, exhibits high sequence recovery in both ProteinMPNN and ESM-IF1. More pronounced differences between ProteinMPNN and ESM-IF1 are observed at CDR3α position 134. In CDR3β, ProteinMPNN achieved the highest sequence recovery at position 109, which usually corresponds to the second serine residue of the CASS motif derived from the V gene. This might suggest some bias from model training.

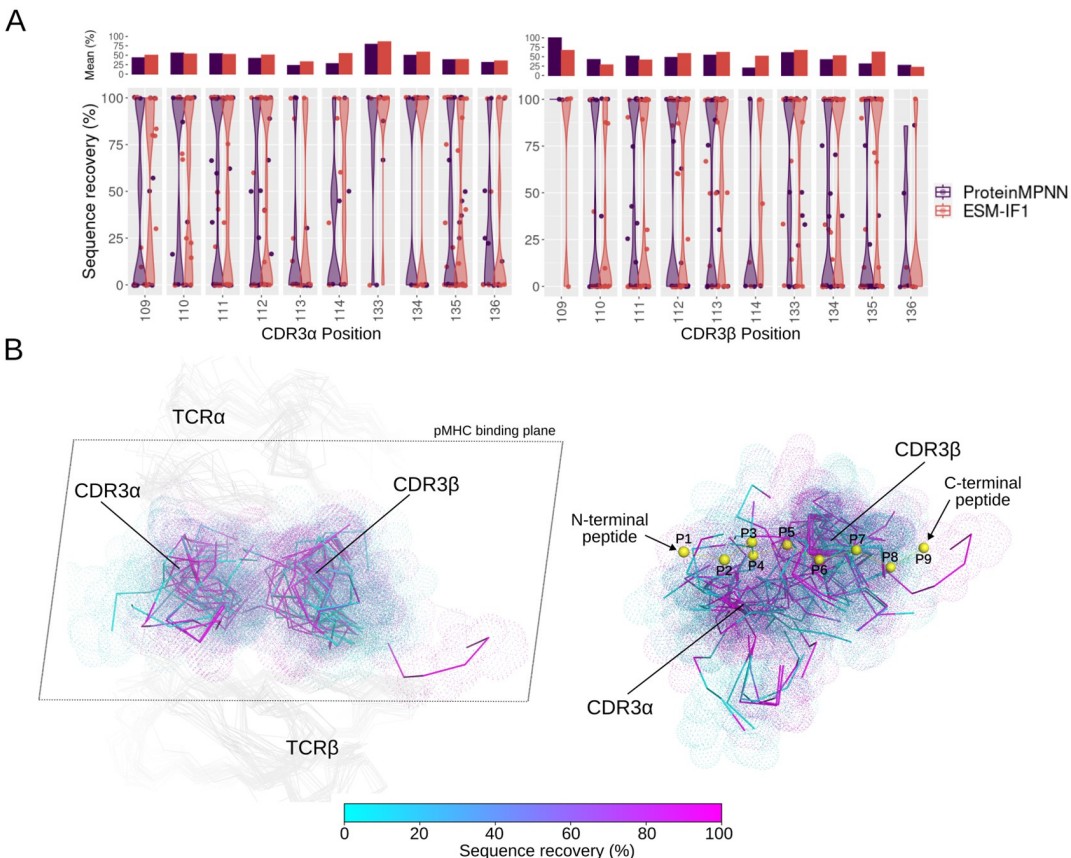

**Fig 2. ProteinMPNN and ESM-IF1 sequence recovery per CDR3 designed positions. (A)** The distribution of the percentage of sequence recovery per designed CDRα (on the left) and CDR3β (on the right) position is depicted as violin plots. Each point represents the average sequence recovery over non-redundant designed amino acids for a given test case at a given position. The position numbering follows the AHO numbering scheme. The upper bar plot shows the average over each sequence recovery distribution. Positions with fewer than three designed cases were removed for clarity. **(B)** ESM-IF1 sequence recovery is mapped onto the TCR structures. On the left, structures of test cases are superposed by the TCR, and only the TCR α and β chains are presented. The structures are oriented towards the pMHC plane. The CDR3β that stands out among the others is the long CDR3β from the test case 7l1d. On the right, the structures are superposed by the MHC, and only the CDR3s are shown. A representative peptide is presented as yellow spheres to highlight the orientation of the CDR3 segments in relation to the pMHC interface.

In CDR3α, position 114 shows low sequence recovery for ProteinMPNN, contrasting with the recovery observed for ESM-IF1. However, a limitation of this analysis is that CDR3 sequence positions can vary substantially in terms of 3D structure positioning. Therefore, we mapped the per-position sequence recovery of ESM-IF1 onto the TCR structures (Fig 2B). Consistent with the distribution analysis, after TCR superposition, we observed balanced sequence recovery over the TCR interface (Fig 2B, left panel). A possible tendency of higher sequence recovery is noted in a region between CDR3β and CDR3α, whereas CDR3 peripheral regions appear to exhibit lower sequence recovery (Fig 2B, left panel). By superposing the TCR:pMHC structures by the MHC and mapping the sequence recovery onto the CDR3, we also observe evenly distributed recovery, suggesting no bias toward an interacting region of the pMHC (Fig 2B, right panel).

Additionally, besides the design of TCR CDR3 interface positions, we explored three other CDR related design scenarios: the design of all CDR3 positions (α and β chains), not limited to those at the pMHC interface, the design of CDR1, CDR2, and CDR3 positions at the interface

with the pMHC and the design of all CDR1, CDR2 and CDR3 positions. When comparing the sequence recovery across the different ProteinMPNN design scenarios, we observed higher recovery rates when designing all CDR3 positions in comparison to the other design scenarios (S11 Fig). This could potentially result from a combination of factors: a bias originating from the conserved N and C termini of the CDR3s across different V and J germlines (S3 Fig), which might be recognized during model training, and the topological characteristics of the CDR3, especially the β chain. In typical TCR binding modes, the CDR3β is often buried at the center of the TCR interface (S2 Fig), making specific contacts with the pMHC and, consequently, more restricted in its amino acid variability during the design. A similar trend was noted in the designs generated by ESM-IF1. It is noteworthy that ESM-IF1 utilizes the UniRef sequence dataset for model training [21]. While full CDR3 sequences are absent in TCR sequences from UniRef, as they are generated through gene recombination, other CDRs that constitute part of the TCR V germline could potentially be included in the ESM-IF1 training set. However, upon analyzing the results from all CDR designs, we did not observe any overestimation of sequence recovery for these cases.

To provide a better understanding of the challenges in designing TCRs, which may also impact interface design scenarios, we expanded our analysis to include the design of the full TCR variable domain (α and β chains) in complex with the pMHC. The average sequence recovery was 47% for both ProteinMPNN and ESM-IF1, similar to that observed in interface design strategies. However, per position sequence recovery analysis (S12A and S12B Fig) reveals that this average is a combination of both lower and higher recovery rates across different TCR positions. Highly conserved TCR framework positions such as 43 and 91 (Aho numbering) and the cysteine (106) and phenylalanine (139) flanking the CDR3 segment (S3 Fig), consistently showed high recovery, contrasting with the more variable positions such as those within the CDRs, which exhibited lower recovery. The trend of sequence recovery along the TCR sequence generally shows a correlation between ProteinMPNN and ESM-IF1 (S12C Fig).

Overall, these findings underscore the capacity of ProteinMPNN and ESM-IF1, utilizing fixed backbone positions, in generating TCR amino acids closely resembling native sequences of TCR binders in comparison to Rosetta Design. We confined the subsequent analyses presented in the following sections to deep learning methods. Additionally, to prevent any potential training information leaks to our test cases, we focused the subsequent analyses on the design of TCR CDR3 interface positions. This scenario is also more relevant since CDRs other than CDR3 can establish additional extensive contacts with the MHC (S1 Fig), thereby increasing the risk of cross-reactivity. When appropriate in subsequent sections, we also evaluated the ProteinMPNN and ESM-IF1 designs of all CDR positions, which can provide useful information in a design scenario when one does not aim to optimize the interface through point mutations but instead is designing a novel CDR interface.

## Recovery of native interface amino acid composition and assessment of naturalness in V(D)J recombination

The preservation of interface amino acid composition is crucial, particularly for the TCRs, as it plays a key role in maintaining binding specificity and leads folding into a loop structure. Here, we compared the composition of the CDR3 interface in native sequences and design sequences (Fig 3A).

Across the CDR3 (α and β chains) interface of all targets, the designed interfaces generated by both ProteinMPNN or ESM-IF1 generally mirrored the amino acid composition of the native interfaces. As expected (S3C Fig), interface CDR3 positions exhibited a strong bias toward glycine, which was recapitulated by both designing methods. Amino acids with low

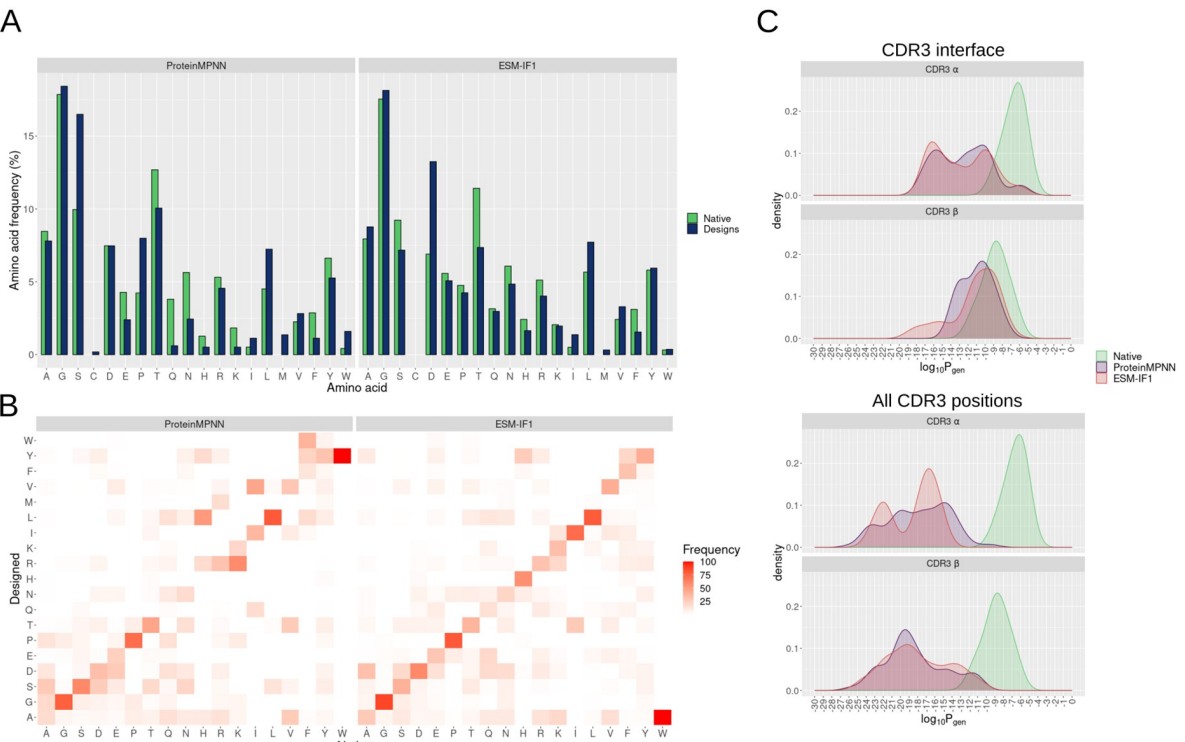

**Fig 3. Occurrence of amino acids at CDR3 interface positions in native and designed sequences. (A)** Frequency of each amino acid at the CDR3 designed positions in native and ProteinMPNN (left panel) or ESM-IF1 (right panel). Higher frequency indicates that the amino acid was more frequently observed at the CDR3 interface in the analyzed test cases. Only non-redundant generated sequences are considered in the analysis. The x axis is ordered by the BLOSUM62 amino acid grouping: [A, G, S], [C], [D, E, P, T], [Q, N, H, R, K], [I, L, M, V], [F, Y, W]. **(B)** Heat map of the frequency of substitutions of the amino acid substitutions in the designed sequences. The x axis represents the amino acids at the native sequences and the y axis represents the corresponding substitution in the designed sequences. A hypothetical frequency of 100% alanine in native sequences and leucine in designed sequence, for instance, indicates that we observed a change from alanine to leucine in all design cases. **(C)** Distribution of the Estimation of Generation Probabilities (Pgen) of CDR3 Sequences Using OLGA [28]. Pgen was estimated for each CDR3 (α or β) generated by ProteinMPNN (in purple) or ESM-IF1 (in red) from different design scenarios (design of only CDR3 interface positions—upper panel—or design of all CDR3 positions—bottom panel). For comparison, Pgen was also estimated for CDR3 sequences from the test case native structures (in green). This analysis considered only designs from human TCR test cases bound to MHC-1, and only Pgen values greater than 0 were presented in the density plot.

occurrence in native CDR3 interfaces, such as cysteine, histidine, isoleucine, lysine, methionine, and tryptophan, were also scarce in the design sequences. However, notable differences were observed, including an increased occurrence of leucine, proline, and especially serine in the ProteinMPNN-designed sequences, along with a reduction in acidic and amide amino acid groups (glutamic acid, glutamine, and asparagine). In ProteinMPNN, some native alanine, threonine, and aspartic acid were substituted by serine, leading to an increased frequency of the latter at the CDR3 interface (Fig 3B). Conversely, the reduction in the occurrence of acidic and amine amino acid groups was not observed in ESM-IF1 which seemed to maintain more balanced substitutions than ProteinMPNN, as expected due to its higher sequence recovery (diagonal trend at the heatmap). Notably, a bias towards aspartic acid was observed in sequences generated by ESM-IF1.

To evaluate the naturalness of the CDR3 designed sequence in terms of V(D)J recombination, we utilized OLGA [28] and estimated the probability (Pgen) of the designed CDR3 being generated by a human natural V(D)J recombination process. Designs of the full CDR3 (α or β) sequence by ProteinMPNN or ESM-IF1 have a probability of generation (Pgen) lower than

$10^{-10}$, suggesting that these sequences would be unlikely generated by the human TCR VDJ recombination process (Fig 3C). A few CDR3β sequences from ProteinMPNN and ESM-IF1, achieved a probability of generation similar to that of CDR3 from native structures. In the scenario where only CDR3 interface positions were designed, the generated sequences achieved a higher probability of generation (Fig 3C). Specifically, 10% of the sequences designed by ProteinMPNN and 16% of those designed by ESM-IF1 had a probability of generation higher than $10^{-10}$, which particularly for CDR3β closely aligns with the distribution of probability generation observed in native structures. This may reflect the substantial native sequence recovery achieved by these methods, and it may also be a consequence of keeping the terminal residues of CDR3, located at VDJ junctions, fixed (not included in the set of designed positions) during the design process. This finding highlights the advantages of using this design strategy when one aims to generate native-like CDR3 sequences.

## Physicochemical related substitutions and sequence recovery at buried or hotspot positions

Since substitutions by physicochemical similar amino acids at protein interfaces can preserve the interface characteristics and, consequently, protein binding, we evaluated the contribution of physicochemical related substitutions to the MHC-I sequence recovery rate by ProteinMPNN and ESM-IF1. Approximately 24% and 18% of the CDR3s interface positions, which underwent amino acids changes in the design process by ProteinMPNN or ESM-IF1, respectively, exhibited substitutions to physicochemical related amino acids (according to BLOSUM62 grouping, see Methods for details). Considering this similarity metric, the averaged sequence recovery increased from 43.9% to 56.9% and from 50.1% to 59.0% for ProteinMPNN and ESM-IF1, respectively (Fig 4).

Another potential factor influencing sequence recovery is that, even within the interface, certain amino acids at specific positions are not entirely buried, thus allowing less restriction in accommodating different amino acids. By focusing exclusively on designed amino acids located at buried interface positions, with a relative solvent accessibility (RSA) lower than 0.2 (as detailed in the methods), we observed a slight but significant average increase to 48.7% in the sequence recovery for ProteinMPNN. However, for ESM-IF1 we did not observe a significant increase in the sequence recovery at buried positions. To better understand the challenges posed by solvent accessibility for these methods, we analyzed the RSA for each CDR3 interface designed amino acid and determined whether the original amino acid at those positions was successfully recovered. Despite the statistically significant differences observed between the recovery and non-recovery groups (S14 Fig), the disparity in RSA ratios is relatively minimal (median of 0.05 for recovered vs. 0.09 for not recovered with ProteinMPNN, and 0.06 for recovered vs. 0.09 for not recovered with ESM-IF1). This analysis suggests that, in general, there is no apparent difficulty for both ProteinMPNN and ESM-IF1 in recovering amino acids at both buried and more solvent-exposed positions at the CDR3 interface with the pMHC. A similar trend was observed when designing all TCRαβ variable domain positions, where the correlation between sequence recovery and RSA per position was moderate to low (S12D and S12E Fig). This indicates that solvent accessibility is not the sole factor impacting recovery.

Given that even buried positions within the interface may not always be critical for the interaction, we identified potential hotspots for pMHC engagement. To achieve this, we conducted computational alanine scanning using Rosetta and identified energetically important designed CDR3 residues for the interaction with the pMHC, considering a 0.5 kcal/mol cutoff as described in the Methods section. Despite observing a significant change for ESM-IF1, overall, we did not clearly observe an effect on evaluating sequence recovery at hotspots positions.

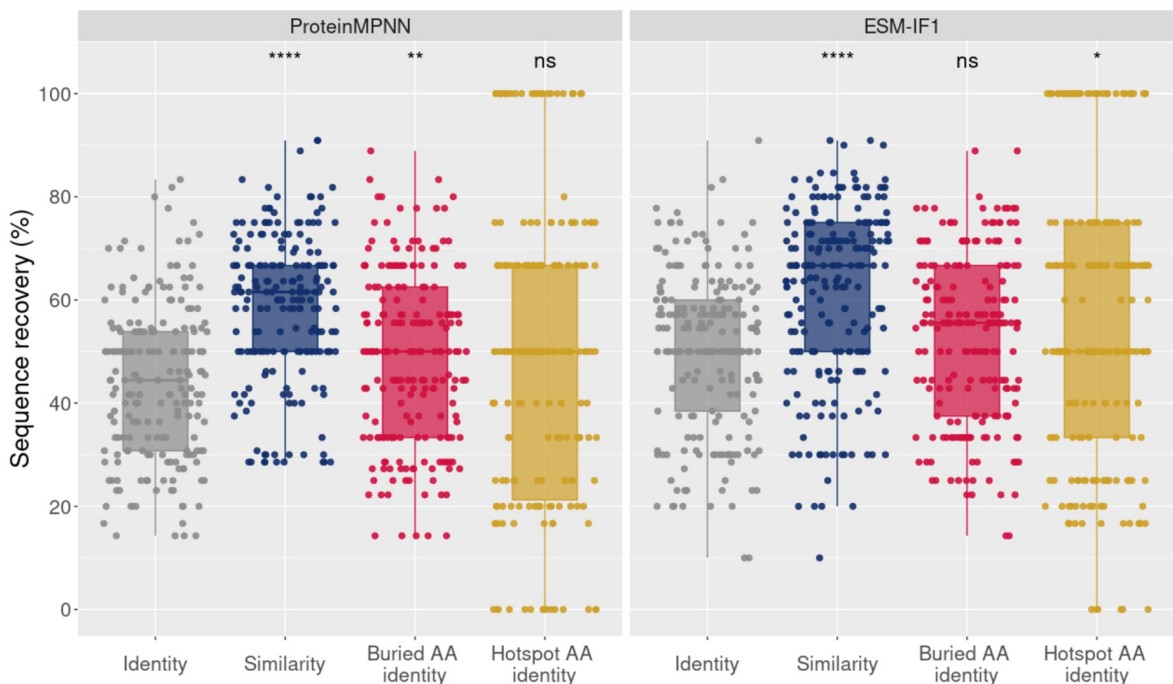

**Fig 4. Sequence recovery analysis of interface CDR3s amino acids in terms of identity, similarity, identity at buried positions and identity at hotspot positions for ProteinMPNN (left panel) and ESM-IF1 (right panel).** For both panels, each point of the box plot represents the percentage of sequence recovery of a unique design sequence from the MHC-I test cases, without redundancy. While identity considers only substitutions to the same native amino acid as recovered, the similarity considers as recovered substitutions to the same amino acid (AA) physicochemical class (see Methods). The buried AA identity corresponds to the identity only computed over buried positions (estimated by relative solvent accessibility, see Methods), whereas the hotspot AA identity corresponds to the identity only computed over interface CDR3s hotspot positions, predicted by computational alanine scanning experiments with Rosetta (see Methods). Statistical pairwise comparison assessed the significance between the identity (reference) and the other metrics. It was performed using the Mann-Whitney test with the R *ggpubr* package. Significance is indicated above each box plot (****, ** and * correspond to a p-value below 0.0001, 0.01 and 0.05, respectively, while 'ns' means no significance (p- value $\geq$ 0.05)). A detailed view of the same evaluated metrics per PDB test case is presented in S13 Fig.

Taken together, these results suggest that the interface recovery by deep learning methods could potentially be even higher when considering physicochemical related substitutions, without being strongly influenced by the solvent exposure of positions or their significance as hotspots for the interaction.

## Designs generated by ProteinMPNN and ESM-IF1 are dependent on the pMHC interface

Despite the notable sequence recovery achieved in deep learning-based designs, there remains a possibility that the model predominantly learns the probability of an amino acid being part of the CDR3 backbone structure or inter/intra TCR chain contacts rather than specifically interacting with the pMHC. To investigate this, we conducted a similar design protocol for designing the CDR3 interface but excluded the pMHC, effectively removing it from the bound TCR:pMHC structure (Fig 5).

As expected, the analysis of maximum sequence recovery, without the pMHC context, revealed a substantial reduction in sequence recovery in 25 (for ProteinMPNN) and 27 (for ESM-IF1) out of 32 cases (Fig 5A). This indicates that, at least for the analyzed TCR:pMHC interfaces, the sequence recovery is not solely a consequence of the TCR backbone structure,

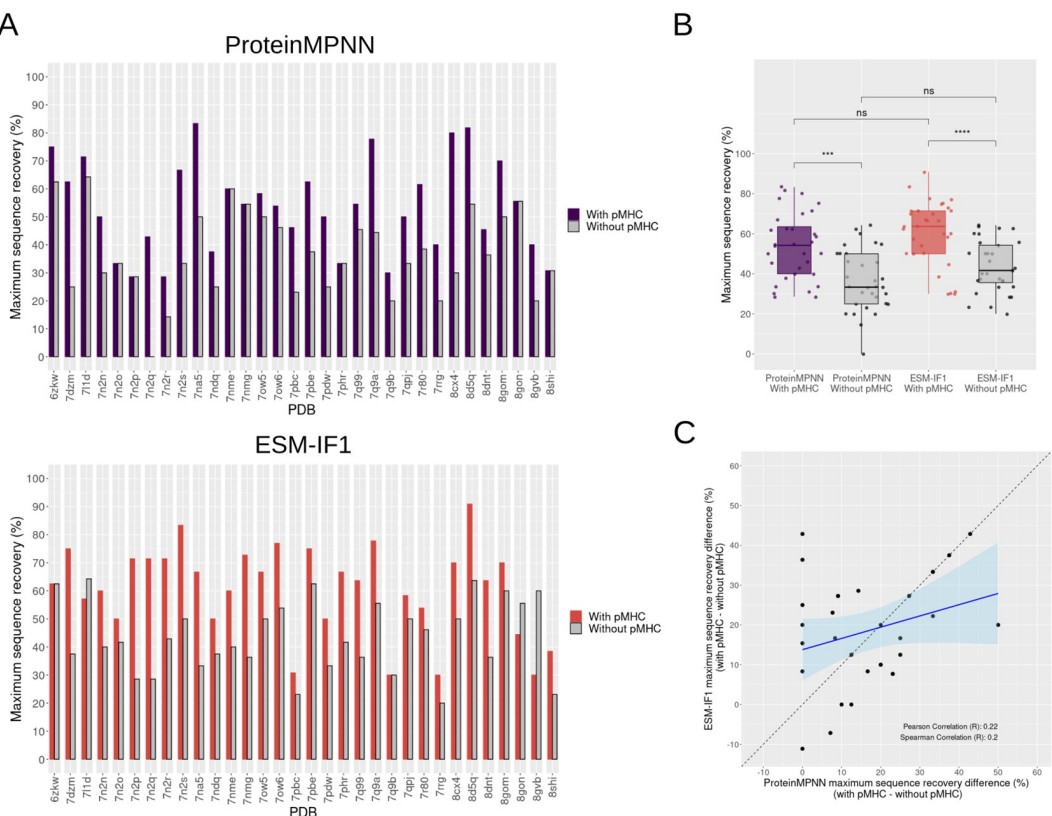

**Fig 5. Maximum sequence recovery of interface CDR3s amino acids in the presence or absence of pMHC structures. (A)** Bar plot displaying the maximum sequence recovery for each test case, with designs generated by ProteinMPNN shown on the top and those by ESM-IF1 on the bottom. Colored bars (purple or red) represent the CDR3 interface designs considering the corresponding pMHC complex, while grey bars represent the interface design of unbound TCRs without the pMHC. **(B)** Same as (A), but grouping together the maximum sequence recovery values for ProteinMPNN with pMHC, ProteinMPNN without pMHC, ESM-IF1 with pMHC, and ESM-IF1 without pMHC. Statistical comparison between groups were performed using Mann-Whitney test with the R *ggpubr* package. Significance is indicated above each box plot (**** and *** correspond to a p-value below 0.0001 and 0.001, respectively, while 'ns' means no significance (p- value ≥ 0.05)). **(C)** Scatter plot with a linear trend line and a 95% confidence interval (light blue region) illustrating the correlation between the difference in maximum sequence recovery upon pMHC removal (maximum sequence recovery with pMHC minus sequence recovery without pMHC) for ProteinMPNN and ESM-IF1. A dashed diagonal line is included to aid in visual comparison. The correlation coefficients are indicated in the plot.

indicating that the models have learned to design the TCR interface when in complex with the target pMHC. In general, we observed a significant reduction on the maximum sequence recovery upon removal of the pMHC around 20%, on average, for both ProteinMPNN and ESM-IF1 (Fig 5B). When correlating the differences observed with and without pMHC per test case between ProteinMPNN and ESM-IF1, we found a low Pearson correlation of 0.22 (Fig 5C). This indicates that the impact on sequence recovery caused by removing the pMHC context is not consistent between ProteinMPNN and ESM-IF1 when analyzed on a per-test-case basis. For instance, in the 7n2p test case, no reduction in maximum sequence recovery was observed for ProteinMPNN, while a 40% reduction was noted for ESM-IF1. Thus, the correlation analysis suggests that these methods can behave differently upon pMHC removal depending on the specific test case. This suggests that the influence of pMHC removal on sequence recovery for each test case is not solely determined by the structure of the TCR: pMHC complex itself, but that the methods may behave differently. Intriguingly, in certain

cases, such as the 7nme for ProteinMPNN and 6zkw for ESM-IF1, the maximum sequence recovery remained unchanged. To determine if the maintenance of maximum sequence recovery with or without pMHC could be due to recovery at non-interface CDR3 positions, we calculated the maximum sequence recovery solely at CDR3 hotspot positions (S15 Fig). Generally, the impact of pMHC removal was more pronounced for both ProteinMPNN and ESM-IF1 when considering only hotspot positions than general interface positions, underscoring the importance of the specific pMHC interface in the design process. In most cases that maintained the same maximum sequence recovery with or without the pMHC when considering general interface positions, the same was true when considering only hotspot positions. This may suggest that for specific cases, additional factors such as the backbone features or TCR inter/intra-chain contacts may play a crucial role in the design process, keeping a high sequence recovery even without the pMHC. Intriguingly, only ESM-IF1 demonstrated higher maximum sequence recovery without the pMHC than with it in three test cases.

## AlphaFold2-based modeling supports the ProteinMPNN and ESM-IF1 TCR designs

In addition to the design evaluation at sequence-level, we performed a structural analysis of TCR designs using a deep learning-based approach. We employed TCRModel2, an Alpha-Fold2 modified version focused on TCR modeling [29]. Design models exhibiting low atomic deviation from the native structure and high modeling confidence score (weighted combination of ipTM and pTM) [29] (S16 Fig) indicate that TCRModel2 modeling supports the interface designed by ProteinMPNN or ESM-IF1, reinforcing our confidence in the generated design. It is important to note that we restricted our analysis to cases where native TCR: pMHC structures from the MHC-I test cases were accurately remodeled with medium or high accuracy (77% of cases) according to CAPRI criteria assessed by DockQ scores. Furthermore, an additional filter was applied based on the TCRModel2 confidence score ($\geq$0.85). When combined, these criteria selected 60% of the cases for use in the structural analysis of designs.

Considering the CDR3 interface design scenario, in 84% and 89% of the analyzed cases, respectively, the ProteinMPNN or ESM-IF1 design with the highest model confidence reached similar or better model confidence than the median of remodeled native sequences. In seven and eight cases, for ProteinMPNN and ESM-IF1, respectively, the designs achieved a model confidence higher than 0.9 (Fig 6A). When randomly selecting dissimilar amino acids (see Methods) at the same positions (excluding cysteine) instead of generating amino acids by ProteinMPNN or ESM-IF1, we can see a clear drop in the model confidence and we observed random designs with the highest model confidence reaching similar or better model confidence than the median of remodeled native sequences in only 48% of the cases (S17A Fig). Additionally, we also observed a higher variability in the model confidence of random designs, with some designs reaching confidence scores below 0.6. It is worth noting that TCR:pMHC interfaces are relatively large and do not involve only CDR3 contacts, thus the occurrence of random designs with high confidence is not unexpected.

We also assessed the deviation between the CDR3 backbone in the designs and the native structure, comparing these deviations to those observed in the remodeled native sequence. Even minimal backbone deviation could be expected due to side chain readjustments required to accommodate designed residues. In 79% of cases for ProteinMPN and 89% for ESM-IF1, at least one design achieved a comparable or lower CDR3 backbone RMSD compared to the median of remodeled native sequence (Fig 6B). Deviations below 2 Å were observed in 53% of cases for both deep learning models. When considering both model confidence and RMSD, in most cases, we observed that the designs, alongside the native sequences, exhibited optimized

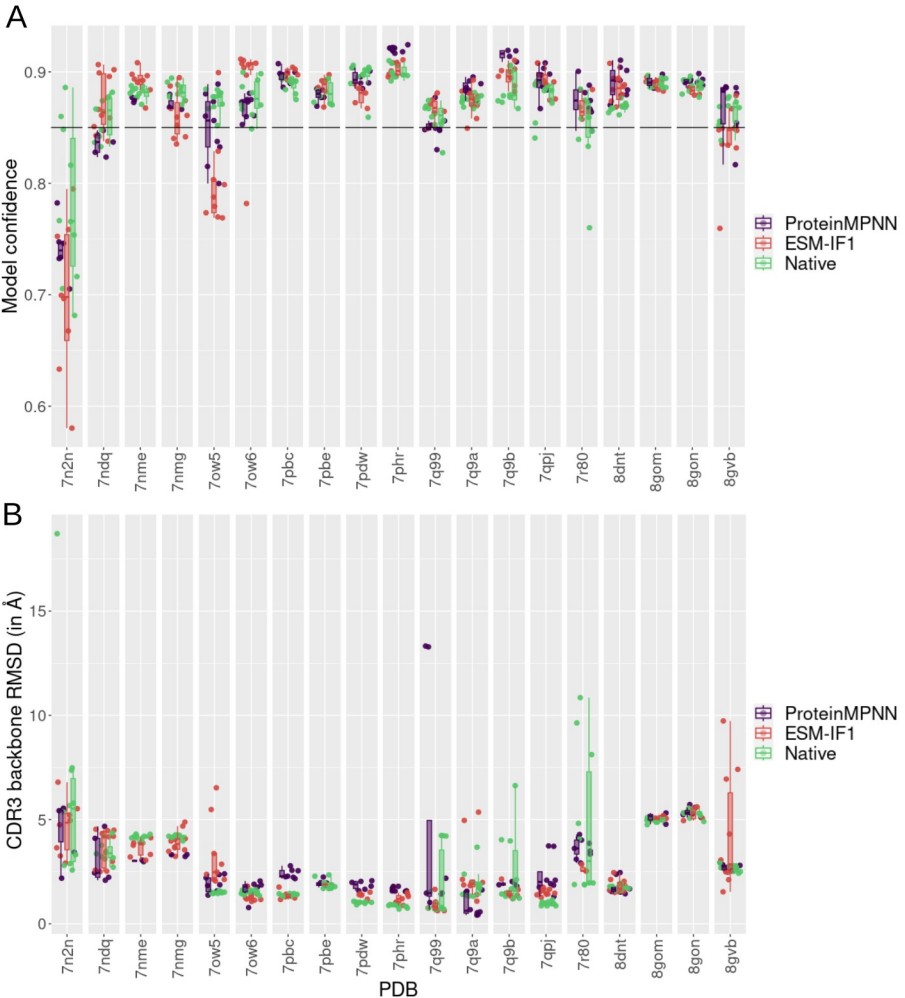

**Fig 6. Modeling of TCR CDR3 interface designs with TCRmodel2. (A)** Box plot of model confidence of ProteinMPNN (in purple) and ESM-IF1 (in red) TCR designs for each test case. Each point represents a different design. The model confidence of remodeled native sequences is colored in green. In this case we remodeled the native sequences 10 times to obtain a distribution of modeling scores that represent the native structure. **(B)** Root-mean-square deviation (RMSD) of CDR3 backbone atoms (both alpha and beta TCR chains) of designs in comparison to the corresponding native crystal structure that originated the designs. RMSDs were determined after structure superposition by the MHC.

model confidence and RMSD (S17C Fig). Similarly to the model confidence analysis, we also observed that random CDR3 interfaces achieved similar backbone RMSD to the designs (S17B Fig), which may be a consequence of other non-designed TCR interface positions stabilizing the complex or obtaining non-disruptive designs by chance. This analysis underscores the importance of utilizing multiple evaluation metrics, as even random designs with low RMSD can present poor confidence.

In addition to the CDR3 interface design scenario, we modeled and evaluated designs from all CDRs positions scenario, where the design of all CDR1, CDR2, and CDR3 positions was permitted (S18 Fig). The model confidence from these designs maintained the high confidence observed in the CDR3 interface designs, with the design achieving the highest model confidence reaching similar or better confidence than the median of remodeled native sequences in

84% and 68% of test cases, respectively for ProteinMPNN and ESM-IF1 (S18A Fig). Confidence scores higher than 0.9 were also observed for both design methods, while no random design achieved a confidence level comparable to that of the native sequences. The high confidence is also consistent with the low CDR deviation from the native structure (RMSD below 2 Å in 79% and 63% of test cases for ProteinMPNN and ESM-IF1, respectively).

Collectively, these findings demonstrate that in the majority of tested cases, the designs generated by ProteinMPNN or ESM-IF1 were successfully modeled by TCRModel2 with high confidence scores, showing relatively low deviation from the CDR native structure. These results suggest that AlphaFold2-based modeling supports the potential binding of most TCR designs, from both the CDR3 interface and all CDR positions design scenarios, to the pMHCs in the test cases.

Additionally, we leveraged the remodeled TCR native complexes to evaluate the performance in terms of sequence recovery of ProteinMPNN and ESM-IF1 on those models instead of the X-ray crystallographic structures (S19 Fig). Generally, we did not observe a strong decrease in maximum sequence recovery of CDR3 interface positions, with an average reduction from 52.1% to 45.3% for ProteinMPNN and from 57.9% to 46.7% for ESM-IF1. For two cases (8gom and 8gon), we observed a substantial reduction in sequence recovery (around 30% to 50%), which may be explained by the deviation between the 3D positions from X-ray and modeled design positions since these two test cases have the highest CDR3 backbone RMSD. The impact caused by 3D deviation and the dependency on backbone positions in the design process can be at least partially observed in the Pearson correlation (0.48 for ProteinMPNN and 0.6 for ESM-IF1). This highlights the importance of using accurate modeling methods for modeling TCR complexes when designing unsolved complexes. Intriguingly, in two analyzed cases (7n2n and 7q9b), we observed an increase in maximum sequence recovery for both ProteinMPNN and ESM-IF1, which may indicate that in these cases the modeling could be improving the quality and placement of the backbone structures.

## Energetic scoring of the TCR designs

As another structure-based criterion to evaluate the TCR designs, we employed the Rosetta energy function to score the designed structures against the corresponding native TCR complexes (the same set used in the AlphaFold modeling assessment). The scoring was based on two widely used Rosetta terms obtained from the Rosetta InterfaceAnalyzer: *dG_separated* and *total_score*. While the former relies on predicted energy changes at the binding interface, the latter provides broader information about complex stability. These two terms can be combined in the design evaluation [56].

The *dG_separated* scoring indicated that both ProteinMPNN and ESM-IF1 CDR3 interface designs followed a similar scoring profile to that determined for the native structures in most test cases (S20A Fig). Designs with worse score, i.e., higher *dG_separated* scores, and higher score difference than native structures were observed in only two test cases (7pdw and 8dnt) for ProteinMPNN. Conversely, ESM-IF1 showed a design with equal or better *dG_separated* score, than the native structure in all test cases. This contrasts with the scores obtained from dissimilar random TCR sequences, which presented higher variability and worse score and higher score differences than native structures in five test cases. This behavior is more pronounced when considering complex stability through the Rosetta *total_score* (S20B Fig). ProteinMPNN and ESM-IF1 designs clearly followed the energetic profile of native structures in all test cases, contrasting with the worse scores, i.e., higher *total_score*, presented by the dissimilar random TCRs. A combination between the two Rosetta terms can be observed in S20C Fig.

When analyzing the scoring of designs covering all CDR positions (CDR1, CDR2, and CDR3), we noted a more pronounced distinction between designs with improved Rosetta scores and those disrupting random interfaces (S21 Fig). In 84% of test cases for ProteinMPNN and 74% for ESM-IF1, we observed designs with improved Rosetta *dG_separated* scores, in contrast to the 31% observed for randomly generated CDR positions. No random TCR achieved a Rosetta *total_score* comparable to that of native or designed TCRs.

## Molecular dynamics simulations and computational binding affinity estimation of the TCR designs

To assess the binding affinity of the TCR designs and compare them with native TCRs, we utilized a robust molecular dynamics simulation protocol coupled with widely used free energy estimation based on a previously benchmarked strategy relying on MM/PBSA calculations [44].

While this MM/PBSA protocol has shown promising results for estimating the binding free energy of two specific TCRs (1G4 and A6), its capacity to generalize to other distinct TCRs remains unexplored. Hence, as a first step to ensure a reliable evaluation of the binding affinity of our TCR designs, we initially benchmarked our protocol to predict free energy changes resulting from mutations in two curated sets of TCR complexes. The first set consisted of wild-type and mutated paired complexes, all with solved 3D structures and experimentally determined binding affinities. The second set, more comparable with the design scenario of this study, included the same wild-type complexes along with mutants lacking solved structures, which were then modeled based on the wild-type structure. The first benchmark test set included 7 different mutants, ranging from single-point to 12 mutations across the TCR α and β chains, while the second set comprised 35 different mutants.

By applying this MM/PBSA protocol, we achieved a Pearson correlation coefficient of 0.9 with experimental data for the set of solved mutant structures (S22A and S22B Fig) and a correlation of 0.74 for the set of modeled mutants (S22C Fig). Additionally, when assessing the MM/PBSA's ability to discriminate improvements in ΔΔG greater than 0.5 kcal/mol or 2 kcal/mol resulting from mutations in the set of modeled mutants, we obtained balanced accuracy scores of 0.79 and 0.85, respectively (S22D Fig). Collectively, these results support the use of MM/PBSA calculations for comparing the binding affinity of native and designed TCRs.

Since this MD-based protocol is time-consuming, we selected CDR interface designs from 7 TCR:pMHC test cases to perform the MM/PBSA calculations over MD simulation trajectories (Fig 7). The trajectories of the designs were stable, maintaining CDR3 backbone deviation medians below 2.5 Å, in most test cases, similarly as the native trajectory (S23 Fig). In general, the Principal Component Analysis (PCA) indicated that the designed CDR3s sampled a similar conformational space as the native ones, with some exceptions like designs from the 7n2o test case (S24 Fig). However, the deviation between these different sampling spaces is low (below 1 Å), as indicated by the RMSD medians (S23 Fig).

Interestingly, in most of the test cases (5 out of 7), we observed designs presenting binding affinity (expressed by ΔG) similar to or better than their corresponding native structure (Figs 7A and S25). Most designs, whether from ProteinMPNN or ESM-IF1, recovered the native binding affinity, presenting no significant difference in comparison to the native ΔG. Nonetheless, in one test case (7q9a) for ProteinMPNN and in three test cases for ESM-IF1 (7n2o, 7q9a and 8dnt), we obtained TCR design with significant higher affinity (lower ΔG) than the native TCR (assessed by a Mann-Whitney test, see S25 Fig). Intriguingly, both methods failed to generate designs with compatible affinity to the native TCR for 7pbe and 7pdw test cases.

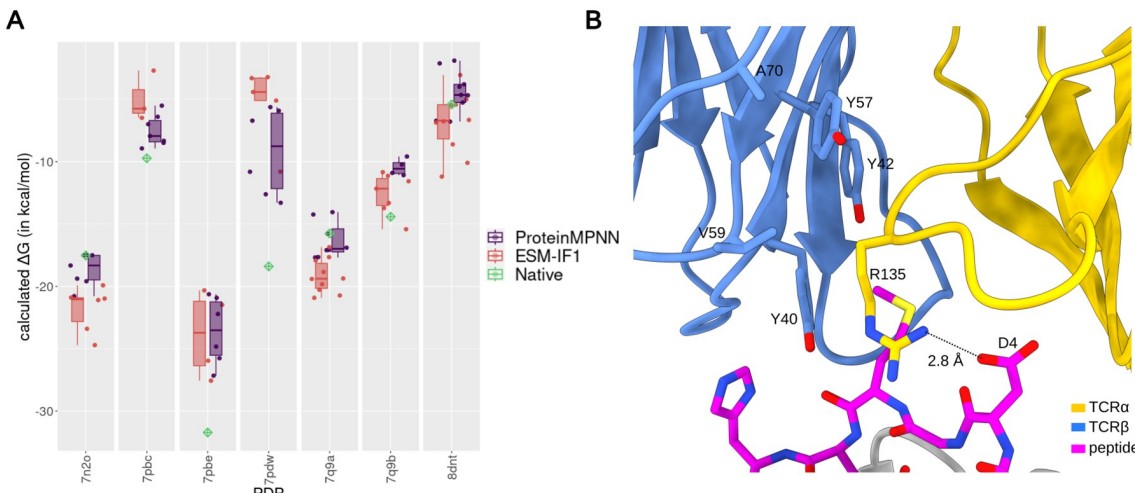

**Fig 7. Molecular dynamics simulation analysis and binding affinity estimation with MM/PBSA of Native and ProteinMPNN and ESM-IF1 TCR designs. (A)** Box plot of calculated ΔG (in kcal/mol) of ProteinMPNN (in purple) and ESM-IF1 (in red) designs in comparison to the calculated ΔG of the native complex (green diamond) for each evaluated MHC-I test case. Each point corresponds to the median of calculated ΔG from 15 replicas. The detailed distribution of ΔG across replicas and statistical tests are presented in S25 Fig. **(B)** Visualization of the TCR and peptide interface from the PDB 7pdw, highlighting the interaction between R135 from the TCRα and the D4 residue of the peptide.

A free energy decomposition of the native 7pdw interaction revealed that most of the TCR interaction relies on TCR three arginine from the TCRα (S26 Fig). Particularly, the one with most contribution (R135) anchors to the D4 peptide using a rotamer conformation that allows this interaction even with a high Cα distance and Cβ atoms pointing in opposite directions. The proposed amino acid by ProteinMPNN is a methionine which could interact with hydrophobic residues of TCRβ like V59, while the proposed amino acids by ESM-IF1 are the glutamine or glutamic acid, which could create a polar interaction and/or hydrogen bonds with a cluster of tyrosine residues of TCRβ (Fig 7B). Consequently, the positioning of this arginine hotspot together with the chemical environment can pose a strong challenge to these deep learning methods based on fixed backbones.

As a final test, we estimated the binding affinity of the ProteinMPNN and ESM-IF1 designs generated when designing all CDR1, CDR2 and CDR3 positions for the 7pdw and 7pbe test cases (S27 Fig). We expected that the design methods might produce improvements in the interaction beyond the CDR3 interaction region, such as towards the MHC, potentially improving binding affinity compared to the native complex. For the 7pdw test case, similar to the CDR3 interface designs, the binding energies remained lower than those of the native for all designs of both ProteinMPNN and ESM-IF1. This may highlight the critical role of the R135, also non-recovered in this tested design strategy, for the TCR interaction. However, for the 7pbe, we observed a significant improvement in binding affinity for one ESM-IF1 design (design #4) and a similar binding affinity as the native complex for three ProteinMPNN designs.

A free energy decomposition of the native 7pbe and the ESM-IF1 design #4 revealed positions in the native TCR, outside the CDR3 loops, that either contributed negatively (such as R40 from CDR1β) or did not substantially contribute at all (such as N27 and A30 from CDR1α, S69 from CDR2α, and N60 from CDR2β) to binding with the pMHC (S28A Fig). Specifically, in the ESM-IF1 design #4, the energy of all these positions was improved, contributing to the overall increase in binding affinity observed for this design. In particular, R40 was

replaced by E40, potentially minimizing repulsive forces with the proximal peptide arginine R5 (S28B Fig). Another example of beneficial modification in this design was the replacement of S69 with D69, enabling anchoring to arginine R131 of the MHC (S28B Fig). In the context of improving affinity, these findings highlight the benefits of expanding the designed positions beyond the CDR3 loops, particularly for the 7pbe test case.

## Discussion

In this study, we investigated the utilization of two recently developed deep learning-based protein design methods, ProteinMPNN and ESM-IF1, for designing TCRs interfaces, across various design scenarios. The primary aim was to evaluate the capabilities of these methods in designing TCRs capable of binding to a specific antigenic peptide presented by the MHC, thereby guiding computational design strategies and impacting on TCR engineering.

Despite the advances in protein language models, the limited quantity and quality of TCR sequence data with known specificity to antigens hinder the direct use of generative models only based on amino acid sequences [57]. As a result, an alternative approach to tackle the computational design of TCRs is to base the design process on the 3D structure of the complex formed by the TCR and the peptide presented by the MHC. Indeed, structure-based design of TCRs is a traditional approach that, thus far, relies solely on parameterized energy functions to predict amino acid substitutions that could enhance TCR interactions [14, 18, 19].

Unlike other immunological interactions, such as those between antibodies and antigens, TCR interactions possess several distinctive characteristics. One of the primary distinctions is the binding affinity, which typically falls within the micromolar range. The therapeutic use of TCRs does not necessarily require optimized affinity, since supraphysiological affinities can even prove detrimental to T cell response [14, 58, 59]. These unique characteristics of TCR interaction underscore the need to explore alternative methods for computational design, strategies that not only rely on numerical energy functions but also have the capability to capture native features of this interaction.

In this study, we designed TCRs to bind antigenic peptides presented by MHCs based on an initial 3D structure of the complex using ProteinMPNN and ESM-IF1 with fixed-backbones. To assess the quality and success of these designs, we compared them to experimentally determined TCRs with solved 3D structures bound to the pMHC, serving as the initial models for the design process. It is worth noting that structure-based design approaches are particularly useful when aiming to enhance an existing interaction or even modify it to minimize cross-reactivity. However, recent advances in generative protein backbone methods, such as RFDiffusion [60], have overcome the limitation imposed by the need for experimentally determined initial structure models, thereby enabling the generation of protein backbones to target a specific protein, which can subsequently be designed using deep learning methods.

The evaluation of designs was based on a comprehensive set of orthogonal computational approaches, primarily utilizing two different kinds of metrics: sequence-based and structure-based metrics. At the sequence-based level, we found that the deep learning methods were able to recover a substantial percentage of native amino acids at the CDR3 interface designed positions, outperforming a physics-based method like Rosetta Design. Notably, when comparing ProteinMPNN and ESM-IF1, we took care to conduct a detailed temperature sampling analysis to ensure comparability for this specific task. Interestingly, in our analyzed test cases, the sequence recovery of ESM-IF1 was generally higher than that of ProteinMPNN. Despite both methods being based on graph neural networks, the training sets differ significantly. While ProteinMPNN is tranied on non-redundant solved 3D structures available in the PDB, comprising approximately 25,000 clusters of heteroligomeric assemblies, ESM-IF1 was trained on

single-chain 3D structures composed by the combination of 12 million modeled UniRef sequences and 16,000 solved structures. Despite not being explicitly trained on oligomeric interfaces, ESM-IF1 was able to recover native sequences, even outperforming ProteinMPNN, which could be attributed to the larger number of training samples. The sequence recovery of ProteinMPNN and ESM-IF1 across the test cases was not strongly correlated, and the designed sequences were not the same, indicating that combining designs from both methods could be beneficial. Interestingly, in the sequence recovery analyses of CDR3 interfaces or even full TCRs, we did not observe a strong correlation between recovery and solvent accessibility, unlike what is observed for general proteins [20]. This highlights other factors particular to these TCR complexes that influence the design. Importantly, the sequence recovery of the CDR3 interface was not significantly affected (only a 10% reduction) when designs were based on TCRModel2-modeled structures instead of experimentally determined structures. However, models with higher CDR3 deviations from the solved structures were unable to generate designs with adequately recovered sequences, underscoring the importance of using accurate modeling approaches when designing unsolved TCR interfaces.

Of note, the sequence recovery obtained in our tests is comparable to that reported in the original proteinMPNN study (51% of recovery for heteromeric interfaces) and ESM study as well [20, 21]. This indicates that despite the peculiarities of TCR interfaces, such as being composed of loops, they did not affect the capability of those methods in recovering the known binder interface. However, this level of sequence recovery was not consistently observed across all analyzed test cases, indicating that some cases may present more challenges. Specific biases observed for general ProteinMPNN designs [20], such as reduction in glutamine incorporation, are also evident in our ProteinMPNN designs. However, in our ProteinMPNN designs we observed a reduced presence of charged amino acids, particularly glutamic acid and lysine, and an increased presence of serine, which contradicts a tendency observed in ProteinMPNN general designs to convert polar residues into charged amino acids [20]. We hypothesize that these differences may be a consequence of the unique characteristics of the TCR:pMHC interface.

When designing all CDR3 positions (either α or β chains) using ProteinMPNN or ESM-IF1, we observed that the probability of these sequences being naturally generated by the human V(D)J recombination process was low. This low probability of generation may stem from the fact that these methods are conditioned to amino acid contacts in 3D space, which are not naturally considered during the nucleotide recombination process. Additionally, the design process in this scenario is tailored to an antigen interface, differing from the antigen-unconditioned TCR recombination process. Conversely, when designing only the CDR3 interface positions, we found higher generation probabilities, closer to those of natural sequences. This could be a result of excluding the terminal positions of CDR3, which are not at the interface, during the design process. Future modifications to these design methods, as proposed in [61] for epitope design, could integrate OLGA and condition the design process to favor the generation of sequences with a higher probability of generation.

Importantly, the estimated sequence recovery in CDR3 interface design was associated with the presence of the pMHC, suggesting a binding-dependent nature, as the same design in the absence of the target pMHC substantially reduced the recovery of native amino acids. This is significant to demonstrate that the predictions, even from a model such as ESM-IF1 that was only trained on single-chain structures, are not solely based on the TCR backbones or even TCR intrachain contacts but are specifically dependent on the target pMHC. Future analyses may also focus on designing stable TCR unbound states in combination with the design of the complex interface, which could contribute to generating improved TCR binders.

Despite sequence recovery being a valuable and widely used metric for assessing the performance of protein design methods, TCRs with low sequence recovery can still exhibit binding.

Indeed, diversity is usually advantageous in a protein design context, particularly for TCR: pMHC interfaces, which may lack naturally optimal interactions [8]. Additionally, it is possible that even with high sequence recovery, a single amino acid divergence compared to the native sequence can have significant effect on the protein stabilization and binding. For this reason, we evaluated the quality of designs of CDR3 interface positions and designs of all CDRs positions using a combination of structure-based metrics. Both AlphaFold2-based modeling and Rosetta scores suggested that the TCR designs generated by ProteinMPNN or ESM-IF1 can form stable complexes with target pMHCs at a similar level to native TCRs.

As a critical characteristic of TCR interfaces, we assessed the binding affinity of the CDR3 interface designed TCRs to pMHC targets using a benchmarked protocol of MD simulations with MM/PBSA for the estimation of the ΔG of binding. Our results suggested that, for most cases, the TCR designs, whether from ProteinMPNN or ESM-IF1, recapitulate the binding affinity of the native TCR. In only two test cases (7pdw and 7pbe), the models failed to generate TCRs with compatible affinity to the native TCRs. These cases, such as the 7pdw case, may serve as study cases for improving the design methods and highlighting the limitations of backbone design methods. Although ESM-IF1 was able to generate enhanced TCRs (as observed in the 8dnt test case), the improved ΔG was not substantial, emphasizing that these methods alone may not consistently optimize the interface. If an optimized interface is required, a combination with an optimization algorithm, as reported in [23], could be beneficial. Interestingly, for the 7pbe test case, designing all CDR positions enabled the generation of a TCR with improved or comparable affinity to the native. Expanding the design beyond the CDR3s may increase the chance of energetically improving TCR positions, particularly those that do not natively contribute positively to binding. Additionally, increasing the number of designed positions could facilitate cooperative residue changes, optimizing the local environment through multiple coordinated amino acid modifications. However, such designs should be evaluated with caution when specificity is crucial, since expanding CDRs beyond the CDR3 may increase nonspecific contacts with the MHC.

In this context, a critical limitation posing challenges to the practical applicability of engineered TCRs is their potential for cross-reactivity, exemplified by the fatal cross-reactivity observed in a clinical study with an affinity-matured TCR that was reactive against the human titin self-peptide [15]. Although cross-reactivity is not the focus of this study and therefore not assessed for the deep learning-generated TCR designs, we believe that the advancements in these protein design methods can contribute to the development of more specific TCRs. These can be further investigated by cross-reactivity predictive methods [62, 63]. Future modifications to ProteinMPNN and ESM-IF1 could, for instance, enable a positive-negative balanced design [64], where TCR positions in contact with the peptide could be optimized to enhance affinity, while positions interfacing with the MHC could be optimized to reduce affinity.

In conclusion, our study demonstrates that deep learning design methods such as ProteinMPNN and ESM-IF1, which utilize fixed-backbone 3D structures of TCR:pMHC complexes, can propose amino acids at the TCR interface with high similarity to known binder sequences, predominantly guided by pMHC contacts. Computational structure-based metrics indicate that the designed TCR complexes are stable, exhibiting a binding affinity similar to native TCRs, with some cases even showing improved affinity. These findings contribute to advancements in the field of TCR design and, consequently, in T cell-based immunotherapy.

## Supporting information

**S1 Fig. Assessment of TCR contacts with the pMHC in the MHC-I and MHC-II test sets.**
**(A)** The upper panels show contacts between the TCRα variable domain and the peptide or

MHC-I, while the lower panels display contacts between the TCRβ variable domain and the peptide or MHC. The left panels detail TCRα and TCRβ contacts with the MHC-I, and the right panels show TCRα and TCRβ contacts with the peptide. Contacts are identified by TCR position, using the Aho numbering scheme, with CDR positions indicated. The y-axis represents the number of structures (total of 32 in the MHC-I test set) where each contact is observed. **(B)** same as (A) but considering contacts with the peptide and MHC in the MHC-II test set (total of 6 structures).
(TIF)

**S2 Fig. Assessment of the Relative Solvent Accessibility (RSA) for each position of the TCRα and TCRβ variable domains. (A)** Box plot of the RSA for TCRα positions (upper panel) and TCRβ positions (lower panel) considering the TCRs from the complex bound with pMHC-I (a total of 32 complexes in the test set). **(B)** same as (A) but considering the TCR from the set of MHC-II bound complexes. CDR positions are indicated.
(TIF)

**S3 Fig. Sequence logos representing the sequence variability across the human TCR variable domain. (A)** Sequence variability across the TRAV domain. The sequences are numbered according to Aho numbering scheme. Sequence gaps consistently observed at the same position in all sequences were removed from the logos for clarity. See the Methods section for details on the construction and processing of the sequence set. **(B)** Same as (A), but for TRBV domain. **(C)** Sequence variability across human CDR3α (on the left) and CDR3β (on the right) sequences from TCRs with known antigen specificity. **(D)** Sequence variability across TRAJ (on the left) and TRBJ (on the right). TRAJ and TRBJ numbering were assigned based on the CDR3 C-terminus as reference.
(TIF)

**S4 Fig. Sequence recovery analysis of interface CDR3 amino acids in designing with ProteinMPNN ESM-IF1 or Rosetta FastDesign (InterfaceDesign2019 protocol). (A)** Upper panel shows the number of interface amino acids selected to be designed for each test case. Bottom panel presents boxplots of sequence recovery for each case designed by ProteinMPNN (purple), ESM-IF1 (red) or Rosetta (blue). Each point corresponds to a design sequence. Redundant designs were removed from the analysis. For both methods, 10 designs were generated per test cases and for Rosetta. Lines indicate the median computed over all designs: 44.4%, 50.0% and 30.8% for ProteinMPNN, ESM-IF1, and Rosetta Design, respectively. **(B)** Same as (A), but for MHC-II. Lines are median over all designs: 33.3%, 33.3%, and 21.8% for ProteinMPNN, ESM-IF1 and Rosetta Design, respectively.
(TIF)

**S5 Fig. Effects of temperature sampling on sequence recovery, entropy and uniqueness in the design of interface CDR3s residues with ProteinMPNN and ESM-IF1.** The upper panel presents the sequence recovery relative to the temperature sampling. Each point corresponds to the average sequence recovery of each MHC-I test case. The middle panel presents the entropy of the designed sequences for each MHC-I test case in function of the temperature sampling. Higher entropy indicates higher diversity in the generated sequences. The entropy was calculated using the R *Bio3D* package as an average of positional entropies. The bottom panel shows the uniqueness of generated sequences in function of temperature. Maximum uniqueness (100%) indicates that all generated sequences for a given test case are different. The tested temperatures ranged from 0.000001 to 5.
(TIF)

**S6 Fig. Sequence recovery analysis of interface CDR3 amino acids in designing with different flavors of Rosetta InterfaceDesign2019 protocol.** Dark blue box plots represent the sequence recovery for each target using the default Rosetta InterfaceDesign2019 protocol and generating 10 designs per target. The medium blue box plots represent the sequence recovery of top 10 designs scored by Rosetta *dG_separated* term from a total of 1000 generated designs. The light blue box plots represent the sequence recovery of top 10 design scores by Rosetta *total_score* term from a total of 1000 generated designs. Each point corresponds to a design sequence. Redundant designs were removed from the analysis. Lines indicate the median computed over all designs: 30.8%, 27.3%, and 30.0% for 10 designs with default Rosetta protocol, top 10 best *dG_separated* score designs and top 10 best *total_score* scored designs, respectively. (TIF)

**S7 Fig. Comparison of CDR3 interface maximum sequence recovery obtained from ProteinMPNN and ESM-IF1 for MHC-I test cases. (A)** Scatter plot with a linear trend line and confidence interval of 0.95 (light blue region) presenting the correlation between the maximum sequence recovery of ProteinMPNN and ESM-IF1. Correlation coefficients are indicated in the plot and each point representing an MHC-I test case. **(B)** Scatter plot comparing the maximum sequence recovery of ProteinMPNN and ESM-IF1 with a diagonal line to aid in visual comparison. All designs were obtained with T = 0.1 for ProteinMPNN and T = 0.2 for ESM-IF1. **(C), (D)** and **(E)** panels present the comparison between test cases where ProteinMPNN achieved a higher maximum sequence recovery (at least 10% higher) than ESM-IF1 (6zkw, 7l1d, 7na5, 7pbc, 7rrg, 8cx4, 8gon, and 8gvb test cases in ProteinMPNN group) and test cases where ESM-IF1 achieved a higher maximum sequence recovery (at least 10% higher) than ProteinMPNN (7dzm, 7n2n, 7n2o, 7n2p, 7n2q, 7n2r, 7n2s, 7ndq, 7nmg, 7ow6, 7pbe, 7phr, and 8dnt test cases in ESM-IF1 group). **(C)** Sequence logos of CDR3α and CDR3β from test cases where ProteinMPNN achieved a higher maximum sequence recovery than ESM-IF1 (ProteinMPNN group) are shown in the first row. In the second row, sequence logos are presented for test cases where ESM-IF1 achieved a higher maximum sequence recovery than ProteinMPNN (ESM-IF1 group). The sequences are numbered according to the Aho numbering scheme. Only non-redundant CDR3 sequences (8 for the ProteinMPNN group and 10 for ESM-IF1 group) were considered in the sequence logo and only positions observed in at least one sequence are presented. **(D)** Box plots showing the length distribution of CDR3α (left) and CDR3β (right) from test cases where ProteinMPNN achieved a higher maximum sequence recovery than ESM-IF1 (in purple) and from test cases where ESM-IF1 outperformed ProteinMPNN (in red). Only non-redundant CDR3 are presented. **(E)** Box plots of the Buried Surface Area (in Å$^2$) (left) and Shape Complementarity (right) of the interface between the TCR and pMHC, obtained from https://tcr3d.ibbr.umd.edu/. These comparisons are between test cases where ProteinMPNN outperformed ESM-IF1 (in purple) and those where ESM-IF1 outperformed ProteinMPNN (in red). In (D) and (E), statistical analyses were performed using Mann-Whitney test with the R *ggpubr* package and 'ns' means no significance (p- value ≥ 0.05). **(F)** Percentage of occurrence of native amino acids at positions recovered exclusively by ProteinMPNN (purple bars) or exclusively by ESM-IF1 (red bars) in test cases from ProteinMPNN or ESM-IF1 groups. Only the sequence with the maximum sequence recovery for each test case, as shown in (B), was considered in this analysis. For example, a taller bar for threonine in the ProteinMPNN results indicates that threonine is a common native amino acid at positions where ProteinMPNN successfully recovered the native amino acids, but ESM-IF1 did not, considering the same test case. **(G)** Percentage of occurrence of native amino acids in the local environment of positions recovered exclusively by ProteinMPNN (purple bars) or exclusively by ESM-IF1 (red bars) in test cases from ProteinMPNN or

ESM-IF1 groups. In this analysis, the local environment is defined as amino acids within a 10 Å distance between backbone atoms. Only the sequence with the maximum sequence recovery for each test case, as shown in (B), was considered. Each point represents a CDR3 interface position exclusively recovered by ProteinMPNN or ESM-IF1, along with the associated percentage of amino acid occurrence in its local environment. For instance, a distribution of points with high percentage values for a given amino acid on the x-axis indicates that this amino acid is enriched in positions exclusively recovered by ProteinMPNN or ESM-IF1. (TIF)

**S8 Fig. Maximum sequence recovery from ProteinMPNN and ESM-IF1 increasing number of designs. (A)** Effect of increasing number of ProteinMPNN generated designs (from 5 to 500 generated designs) with T = 0.1 on the maximum sequence recovery for each MHC-I test case. **(B)** Box plot of the maximum sequence recovery per number of generated sequences by ProteinMPNN. Each point corresponds to the maximum sequence recovery observed at the given number of sampled sequences of a given test case. **(C)** Same as (A), but with ESM-IF1. **(D)** Same as (B), but with ESM-IF1. Statistical pairwise comparison between sampling of 10 and 500 using Mann-Whitney test with the R *ggpubr* package. Significance is indicated above each box plot (* corresponds to a p- value $\geq$ 0.05 and, while 'ns' means no significance (p- value > 0.5)). (TIF)

**S9 Fig. Correlation between ProteinMPNN sequence recovery and score. (A)** Scatter plot of sequence recovery and ProteinMPNN scores from 1000 ProteinMPNN designs for each test case. Redundant sequences were not removed in this analysis since the same sequence can have different ProteinMPNN scores. **(B)** Scatter plot considering all designs together with a linear trend line (blue line) presenting the correlation between the sequence recovery and ProteinMPNN score. Correlation coefficients are indicated in the plot. The correlation coefficient was determined using Spearman. (TIF)

**S10 Fig. Comparison of the sequence recovery from 10 ProteinMPNN designs or a top 10 of designs ranked by ProteinMPNN score from a total of 1000 generated designs.** The box plots present the sequence recovery for the 10 ProteinMPNN designs (in dark purple) and for the top 10 scored designs (in light purple). Each point corresponds to the sequence recovery of a designed sequence. Redundant sequences within each test case were excluded from the analysis. Lines indicate the median computed over all designs: 44.4% for both cases. (TIF)

**S11 Fig. ProteinMPNN (A) and ESM-IF1 (B) TCR design considering different design scenarios.** The box plots present the sequence recovery for each design strategy: design of only CDR3 (α and β chains) at the interface with the pMHC (in grey), design of all CDR3 (α and β chains) positions (in blue), design of CDR1, CDR2, and CDR3 positions (α and β chains) at the interface with the pMHC (in pink) and design of all CDR1, CDR2, and CDR3 positions (α and β chains) (in yellow). Each point corresponds to the sequence recovery of a designed sequence. Redundant sequences within each test case were excluded from the analysis. In (A), lines indicate the median computed over all designs: 44.4%, 53.1%, 39.0% and 44.9% for CDR3s interface positions, all CDR3 positions, all CDRs interface positions and all CDRs positions, respectively. In (B), lines indicate the median computed over all designs: 50.0%, 53.8%, 45.3% and 46.9% for CDR3s interface positions, all CDR3 positions, all CDR interface positions and all CDRs positions, respectively. (TIF)

**S12 Fig. Sequence recovery across the entire TCRαβ variable domain for ProteinMPNN and ESM-IF1 designs. (A)** Bar plots showing the averaged sequence recovery per position in the TCRα variable domain for ProteinMPNN (upper, purple) and ESM-IF1 (lower, red). This recovery considers all test cases and the designed sequences from these cases combined. **(B)** Identical to (A) but for the TCRβ variable domain. **(C)** A scatter plot with a linear trend line and a 95% confidence interval (light blue region) illustrates the correlation between sequence recoveries per position from ProteinMPNN and ESM-IF1 for the entire TCR variable domain design scenario. Each point represents a TCRαβ variable domain position, with correlation coefficients detailed on the plot. **(D)** Scatter plot with a linear trend line and a 95% confidence interval (light blue region) shows the correlation between sequence recovery per position from ProteinMPNN and the Relative Solvent Accessibility (RSA) for each position, with correlation coefficients detailed on the plot. **(E)** Similar to (D), this plot correlates RSA with ESM-IF1 sequence recovery per position.
(TIF)

**S13 Fig. Sequence recovery analysis of interface CDR3 amino acids in terms of identity (same amino acids in design and native sequence), similarity (amino acids in design within the same physicochemical group as the one in native sequence), identity at only interface buried positions and identity at hotspot positions for ProteinMPNN (A) or ESM-IF1 (B).** The sequence recovery (in %) is presented by box plots for each case designed by ProteinMPNN or ESM-IF1. Each point corresponds to a design sequence.
(TIF)

**S14 Fig. Relative Solvent Accessibility (RSA) at designed CDR3 interface positions.** Each point represents a position that was either recovered or not during the design process with ProteinMPNN or ESM-IF1, along with the corresponding RSA. Statistical pairwise comparison assessed the significance between the identity (reference) and the other metrics. It was performed using the Mann-Whitney test with the R *ggpubr* package. Significance is indicated above each box plot (****, ** and * correspond to a p-value below 0.0001, 0.01 and 0.05, respectively, while 'ns' means no significance (p- value $\geq$ 0.05)).
(TIF)

**S15 Fig. Maximum sequence recovery of hotspot CDR3s amino acids in the presence or absence of pMHC structures. (A)** Bar plot displaying the maximum sequence recovery for each test case, with designs generated by ProteinMPNN shown on the top and those by ESM-IF1 on the bottom. Colored bars (purple or red) represent the CDR3 interface designs considering the corresponding pMHC complex, while grey bars represent the interface design of unbound TCRs without the pMHC. Hotspot positions were predicted by computational alanine scanning experiments (see Methods) and an asterisk (*) indicates test cases that have fewer than four hotspot positions. **(B)** Same as (A), but grouping together the maximum sequence recovery values for ProteinMPNN with pMHC, ProteinMPNN without pMHC, ESM-IF1 with pMHC, and ESM-IF1 without pMHC. Statistical comparison between groups were performed using Mann-Whitney test with the R *ggpubr* package. Significance is indicated above each box plot (**** corresponds to a p-value below 0.0001, while 'ns' means no significance). **(C)** Scatter plot with a linear trend line and a 95% confidence interval (light blue region) illustrating the correlation between the difference in maximum sequence recovery upon pMHC removal (maximum sequence recovery with pMHC minus sequence recovery without pMHC) for ProteinMPNN and ESM-IF1. A dashed diagonal line is included to aid in visual comparison. The correlation coefficients are indicated in the plot.
(TIF)

**S16 Fig. Correlation between model confidence and DockQ obtained for TCRModel2 models (5 models per each test case) of remodeled native sequences.**
(TIF)

**S17 Fig. Modeling of ProteinMPNN and ESM-IF1 TCR CDR3 interface designs, native and random dissimilar sequences with TCRmodel2. (A)** Model confidence of ProteinMPNN (in purple), ESM-IF1 (in red) TCR designs, remodeled native structures (in green) and random dimissilar sequences (in cyan) for each test case. The secondary upper panel presents the maximum model confidence for each method and test case. **(B)** RMSD of CDR3 backbone atoms (both alpha and beta TCR chains) of designs and random in comparison to the corresponding native crystal structure that originated the designs. RMSDs were determined after structure superposition by the MHC. The secondary upper panel presents the minimum RMSD for each method and test case. **(C)** Scatter plot of the CDR3 backbone RMSD with the model confidence for ProteinMPNN and ESM-IF1 designs, random dissimilar sequences and native sequences. Since random sequences generate models with high deviation, for clarity only RMSD below 15 Å are presented.
(TIF)

**S18 Fig. Modeling of ProteinMPNN and ESM-IF1 TCR designs of all CDRs positions, native and random dissimilar sequences with TCRmodel2. (A)** Model confidence of ProteinMPNN (in purple), ESM-IF1 (in red) TCR designs, remodeled native structures (in green) and random dissimilar sequences (in cyan) for each test case. The secondary upper panel presents the maximum model confidence for each method and test case. Random dissimilar sequences from test cases 7ndq, 7q99 and 7q9a were not able to be modeled by TCRModel2. **(B)** RMSD of CDRs backbone atoms (both alpha and beta TCR chains) of designs and random in comparison to the corresponding native crystal structure that originated the designs. RMSDs were determined after structure superposition by the MHC. The secondary upper panel presents the minimum RMSD for each method and test case. **(C)** Scatter plot of the CDRs backbone RMSD with the model confidence for ProteinMPNN and ESM-IF1 designs, random dissimilar sequences and native sequences.
(TIF)

**S19 Fig. Maximum sequence recovery of CDR3 interface positions from X-ray crystallographic and modeled TCR:pMHC structures. (A)** Maximum sequence recovery (%) for each test case when designing sequences based on X-ray structures (purple bars) or structured modeled with TCRModel2 (grey bars). To obtain the maximum sequence recovery of modeled structures, we modeled the sequence of each test case 10 times with TCRModel2 and selected the top ranked model with the highest model confidence for each test case. Purple (for X-ray) and black (for modeled) dashed lines indicate the median of maximum sequence recovery of 53.8% and 50.0%, respectively, considering all test cases. **(B)** Scatter plot with a linear trend line and confidence interval of 0.95 (light blue region) presenting the correlation between the CDR3 backbone RMSD and the absolute difference in maximum sequence recovery from designs based on X-ray and modeled structures. The CDR3 backbone RMSD measures the deviation in the 3D positions of the CDR3 backbone between the X-ray and modeled structures. Correlation coefficients are detailed in the plot. **(C)** and **(D)** follow the same format as (A) and (B) but focus on ESM-IF1 designs. The median maximum sequence recovery values for ESM-IF1 are indicated by dashed lines, with 60.0% for X-ray and 46.1% for modeled structures.
(TIF)

**S20 Fig. Scoring of ProteinMPNN and ESM-IF1 designs of CDR3 interface positions, native structure and dissimilar random generated structures with Rosetta energy functions. (A)** Rosetta *dG_separated* term obtained from Rosetta InterfaceAnalyzer. Box plots represent *dG_separated* of the ESM-IF1 (in green) and ProteinMPNN (in purple) designs, dissimilar random generated TCRs (in cyan) and the red diamond corresponds to the *dG_separated* of the native structure The secondary upper panel presents the minimum Rosetta *dG_separated* for each method and test case. **(B)** Same as (A), but showing the Rosetta *total_score* term instead of *dG_separated*. The secondary upper panel presents the minimum Rosetta *total_score* for each method and test case. **(C)** Scatter plot presenting the relation between the *dG_separated* and *total_score* for native, designs and random sequences for all test cases. The lower the *dG_separated* and *total_score* scores, the higher the affinity and stabilization, respectively.
(TIF)

**S21 Fig. Scoring of ProteinMPNN and ESM-IF1 designs of all CDRs positions, native structure and dissimilar random generated structures with Rosetta energy functions. (A)** Rosetta *dG_separated* term obtained from Rosetta InterfaceAnalyzer. Box plots represent *dG_separated* of the ESM-IF1 (in green) and ProteinMPNN (in purple) designs, dissimilar random generated TCRs (in cyan) and the red diamond corresponds to the *dG_separated* of the native structure. The secondary upper panel presents the minimum Rosetta *dG_separated* for each method and test case. **(B)** Same as (A), but showing the Rosetta *total_score* term instead of *dG_separated*. The secondary upper panel presents the minimum *total_score* for each method and test case. **(C)** Scatter plot presenting the relation between the *dG_separated* and *total_score* for native, designs and random sequences for all test cases. The lower the *dG_separated* and *total_score* scores, the higher the affinity and stabilization, respectively.
(TIF)

**S22 Fig. Binding affinity estimation of wild-type and mutant TCR complexes using molecular dynamics simulation and MM/PBSA calculations. (A)** Box plots of calculated $\Delta G$ (in kcal/mol) for the wild-type TCR complex (wt, in blue) or mutant complex (mut, in red) from MM/PBSA. Each point corresponds to a replica (15 in total) of a molecular dynamics simulation trajectory. The values of the experimental and calculated ($\Delta\Delta G_{mut-wt}$) (in kcal/mol) are shown above the box plot panel. **(B)** Scatter plot with a linear trend line and confidence interval of 0.95 (light blue region) presenting the correlation between the experimental and calculated $\Delta\Delta G$ considering the set of solved wild-type and mutant structures. Correlation coefficients are indicated in the plot. **(C)** Same as (B) but considering the set of modeled mutant structures. **(D)** Confusion matrix created from the scatter plot presented in (C) presenting the balanced accuracy and F1 score in the discrimination of $\Delta\Delta G$ greater than -0.5 kcal/mol (upper panel) or -2 kcal/mol (bottom panel).
(TIF)

**S23 Fig. RMSD of CDR3 (α and β chains) backbone estimated from trajectories of molecular dynamics simulations of ProteinMPNN and ESM-IF1 designs, as well as the native trajectory.** The RMSD values, depicted as box plots, were calculated subsequent to superposing the trajectory frames by the MHC of the corresponding reference crystal structure. For each test case, all replicas of all designs were combined to form a single RMSD distribution.
(TIF)

**S24 Fig. Principal Component Analysis (PCA) of molecular dynamics simulations of ProteinMPNN and ESM-IF1 designed TCRs and native TCRs.** For the PCA analysis, only the CDR3 coordinates were considered (see Methods). The first two main components are

represented as 2D density contours (bins of 50) colored by R viridis scale that ranges from dark purple to yellow, being yellow the regions of higher density. Plots were built using R *ggplot* package.
(TIF)

**S25 Fig. Binding affinity estimation of ProteinMPNN (A) and ESM-IF1 (B) TCR designs by MM/PBSA calculations.** Each plot corresponds to a TCR:pMHC test case. The box plots present the ΔG (in kcal/mol) calculated for each of the 15 replicas of the TCR designs (ProteinMPNN in purple and ESM-IF1 in red) and the wild-type (wt) TCR (in green). The TCR designs are presented by IDs with a maximum of 10 designs. The lower number of designs are a consequence of redundant generated designs that were removed for the calculations. The statistical difference between each design and the corresponding wild-type was determined by Mann-Whitney test and the significance is indicated above each box plot (***, ** and * correspond to a p-value below 0.001, 0.01, and 0.05, respectively, while 'ns' means no significance).
(TIF)

**S26 Fig. MM/PBSA decomposition (in terms of ΔG, in kcal/mol) of the TCR residues included in the design of the 7pdw test case.** Residues from TCRα (green) and TCRβ (red) are shown. Low ΔG values indicate the residue contributed positively to the binding energy and high ΔG values indicate the residue contributed negatively to the binding energy.
(TIF)

**S27 Fig. Binding affinity estimation with MM/PBSA of ProteinMPNN (A and C) and ESM-IF1 (B and D) TCR designs from the all CDR positions strategy for 7pdw (A and B) and 7pbe (C and D) test case.** The box plots present the ΔG (in kcal/mol) calculated for each of the 15 replicas of the TCR designs (purple for ProteinMPNN and red for ESM-IF1) and the native TCR (in green). The TCR designs are presented by IDs with a maximum of 10 designs. The statistical difference between each design and the corresponding wild-type was determined by Mann-Whitney test and the significance is indicated above each box plot (***, ** and * correspond to a p-value below 0.001, 0.01, and 0.05, respectively, while 'ns' means no significance).
(TIF)

**S28 Fig. MM/PBSA decomposition of all CDR residues for the native 7pbe complex and the ESM-IF1 design #4. (A)** Free energy decomposition (in kcal/mol) per CDR position for the native 7pbe complex (green) and the ESM-IF1 design #4 (red). Panels are split by CDR regions. Low ΔG values indicate residues that contributed positively to the binding energy, while high ΔG values indicate residues that contributed negatively. **(B)** Atomic contacts observed in the native 7pbe complex (first column panels) and in the ESM-IF1 design #4 (second column panels). The TCRα chain is shown in yellow, the TCRβ chain in blue, the peptide in magenta, and the MHC in grey. Residues involved in the highlighted contacts are represented as sticks, with atomic distances indicated. The atomic structures presented were obtained through energy minimization.
(TIF)

**S1 Appendix. Rosetta resfile example.**
(PDF)

**S2 Appendix. Rosetta design protocol.**
(PDF)

**S3 Appendix. Rosetta2.3 command line for alanine scanning.**
(PDF)

**S4 Appendix. Example of mutation list file for Rosetta2.3 alanine scanning protocol.**
(PDF)

**S5 Appendix. Rosetta interface analyzer protocol.**
(PDF)

**S1 Table. List of PDB IDs of MHC-I and MHC-II TCR:pMHC complexes used for comparing design methods.**
(PDF)

**S2 Table. List of the wild-type and the corresponding mutant PDB structures of TCR: pMHC complexes that compose the benchmark for binding affinity calculations with MM/ PBSA.** The table includes the experimental ΔG (in kcal/mol) for each complex and the corresponding ΔΔG ($\Delta G_{mut}$—$\Delta G_{wt}$). All experimental data was collected from ATLAS database (https://atlas.wenglab.org/).
(PDF)

**S3 Table. List of the wild-type and the corresponding TCR mutant that composes the benchmark of modeled mutants for binding affinity calculations with MM/PBSA.** The table includes the experimental ΔG (in kcal/mol) for both wild-type and mutant, as well as the corresponding ΔΔG ($\Delta G_{mut}$—$\Delta G_{wt}$). All experimental data was collected from ATLAS database (https://atlas.wenglab.org/). (*) The mutation numbering presented follows the same used in ATLAS database.
(PDF)

## Acknowledgments

We are grateful to the Brazilian Biosciences National Laboratory (LNBio), part of the Brazilian Center for Research in Energy and Materials (CNPEM) for accessibility to the High-Performance Computing Cluster and scientific infrastructure. We thank the ProteinMPNN, ESM-IF1 and Rosetta teams for sharing their algorithm and code.

## Author Contributions

**Conceptualization:** Helder V. Ribeiro-Filho, Brian G. Pierce, Paulo S. Lopes-de-Oliveira.

**Data curation:** Helder V. Ribeiro-Filho, Gabriel E. Jara, Melyssa Cheung, Nathaniel R. Felbinger.

**Formal analysis:** Helder V. Ribeiro-Filho, Gabriel E. Jara, João V. S. Guerra, Melyssa Cheung, Nathaniel R. Felbinger, José G. C. Pereira, Brian G. Pierce, Paulo S. Lopes-de-Oliveira.

**Funding acquisition:** Helder V. Ribeiro-Filho, Brian G. Pierce, Paulo S. Lopes-de-Oliveira.

**Investigation:** Helder V. Ribeiro-Filho, Gabriel E. Jara, João V. S. Guerra, José G. C. Pereira, Brian G. Pierce, Paulo S. Lopes-de-Oliveira.

**Methodology:** Helder V. Ribeiro-Filho, Gabriel E. Jara, João V. S. Guerra, José G. C. Pereira, Brian G. Pierce, Paulo S. Lopes-de-Oliveira.

**Project administration:** Helder V. Ribeiro-Filho, Brian G. Pierce, Paulo S. Lopes-de-Oliveira.

**Supervision:** Helder V. Ribeiro-Filho, Brian G. Pierce, Paulo S. Lopes-de-Oliveira.

**Validation:** Helder V. Ribeiro-Filho, Gabriel E. Jara, João V. S. Guerra, Brian G. Pierce.

**Visualization:** Helder V. Ribeiro-Filho, Gabriel E. Jara.

**Writing – original draft:** Helder V. Ribeiro-Filho, Gabriel E. Jara, João V. S. Guerra, Melyssa Cheung, Nathaniel R. Felbinger, José G. C. Pereira, Brian G. Pierce, Paulo S. Lopes-de-Oliveira.

**Writing – review & editing:** Helder V. Ribeiro-Filho, Gabriel E. Jara, João V. S. Guerra, Melyssa Cheung, Nathaniel R. Felbinger, José G. C. Pereira, Brian G. Pierce, Paulo S. Lopes-de-Oliveira.

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
