## [Decision Letter · Decision Letter 0]

14 Jun 2024

Dear Dr Ribeiro,

Thank you very much for submitting your manuscript "Exploring the Potential of Structure-Based Deep Learning Approaches for T cell Receptor Design" for consideration at PLOS Computational Biology.

As with all papers reviewed by the journal, your manuscript was reviewed by members of the editorial board and by several independent reviewers. In light of the reviews (below this email), we would like to invite the resubmission of a significantly-revised version that takes into account the reviewers' comments.

We cannot make any decision about publication until we have seen the revised manuscript and your response to the reviewers' comments. Your revised manuscript is also likely to be sent to reviewers for further evaluation.

Sincerely,

Dina Schneidman

Academic Editor

PLOS Computational Biology

Nir Ben-Tal

Section Editor

PLOS Computational Biology

Reviewer's Responses to Questions

**Comments to the Authors:**

Reviewer #1: In this work, the authors evaluate different inverse folding algorithms for TCR-pMHC binding design.

This is great and thorough work. i have a few comments.

- you use sequence recovery as a main metric. Given that one wants diversity in the designed sequences, why is sequence recovery, which measures the conservation of native positions, a good metric? can you explain this better in the manuscript?

- can you also use immunebuilder (in additon to tcrmodeler2) to investigate the designs structurally?

- can you also run your designed TCRs through OLGA and test if they are actually actually native-like TCRs? If the sequence pgen is very low, then the designs are likely not very natural and thus not useful for therapeutics application (immunogenic).

Reviewer #2: In this manuscript, Ribeiro-Filho et al. explore the application of inverse folding models (ProteinMPNN and ESM-IF1) to the fixed-backbone CDR sequence design of TCRs. They evaluate the models’ performance via sequence recovery, refolding, and energetic scoring (using Rosetta and MM/PBSA calculations). The authors demonstrate that the tested inverse folding methods can recapitulate a substantial portion of native sequences and produce designs with equal, and in some cases greater, predicted binding affinity to the target peptide-MHC complex than the starting TCR sequence.

This work provides value to the field by examining the performance of inverse folding models on TCRs, which, as the authors point out, are underexplored in the protein design space. However, the manuscript is currently somewhat limited in novelty and scope. The authors use existing, well-established inverse folding models and evaluation methods. Additionally, the sequence design analysis is primarily limited to CDR3 loops. The paper would be strengthened by addressing the following points.

Major comments

• The manuscript should make a stronger case for the intended application(s) and the applicability (or lack thereof) of the inverse folding models. As it stands, only the re-design of CDR (primarily CDR3) loops for structurally-resolved TCR-pMHC complexes is investigated. This may only cover a fraction of real-world use cases for TCR inverse folding. Extending the analyses to some of the following suggestions would provide a more thorough understanding of the models’ applications and limitations:

– What are the inverse folding models stronger or weaker at, respectively and/or in general? A more detailed investigation of failure modes would be helpful.

– Expand sequence design beyond the CDR3. Given that the randomly designed CDR3 sequences had similar RMSDs to the designed sequences and TCRs with the same CDR3 sequence can bind different pMHC complexes (as mentioned in section 4.1), an expanded analysis of sequence design for other TCR regions could provide more insights into model performance. Additionally, this would open the door to further important evaluations, such as whether generated sequences map back to known V/J genes (and, if so, whether these genes are associated with the native loop canonical forms).

– Apply the inverse folding approaches to predicted structure inputs. In real design use cases, one may not always have a solved structure of the starting complex. Are the models still able to achieve strong performance without an experimentally solved structure?

– Polyreactivity: although the authors state this is beyond the scope of their manuscript, it would be valuable to understand, for example, if the inverse folding model suggestions increase affinity for the MHC rather than the peptide and thereby increase the risk of polyreactivity.

• The MM-PBSA affinity calculation benchmark gives a helpful understanding of the potential accuracy of this approach. However, this accuracy is not directly applicable to the use case in this manuscript: for the benchmark, only cases with a solved structure of the wild-type and mutant were included; for the assessment of inverse folding designs, the mutant structures were modeled. The MM-PBSA benchmark should be updated/extended to include cases with modeled mutant structures to provide a relevant quantification of this protocol’s accuracy.

• It would be useful to provide an understanding of natural sequence variability in TCRs across different portions of the structure (down to single-position resolution for CDR3 at least). This will give more insights into the relative challenges of sequence recovery tasks.

Minor comments

• It would be helpful if the authors more clearly stated/explained the following:

– In section 2.1, it should be more clearly stated that the antigen is provided as context for the models. Additionally, the authors should explain whether the rest of the TCR sequence is given to the model as context and whether the CDR sequences were sampled autoregressively or in a single forward pass.

– In Figure 1B, a sequence recovery value of exactly 50% appears to be overrepresented. Is there an explanation for this?

– What percent of model designs are redundant?

– In the Figure 4 legend, the buried AAs are defined as a subset of the hotspot AAs, but this does not seem to match section 4.5.

– In section 2.5, the TCRModel2 confidence metric should be explained, as this is a major underpinning of the analyses presented in this section. Additionally, it should be clearly specified which confidence metric is used. (I presume the linear combination of pTM and ipTM, as this is defined as ‘model confidence’ in the TCRModel2 publication, but the paper also states that pLDDT, pTM, and ipTM confidence metrics are provided).

– Also in section 2.5, the authors should state and comment on the results in Figure S10, that random CDR3 sequences achieve similar RMSDs to designed sequences.

– Could the authors elaborate on how the same sequence can achieve different ProteinMPNN scores, as stated in the Figure S5 caption? Would the log-probability outputs not be fixed for a given structure input (and used to sample/score multiple sequence outputs)?

• The following text could be updated:

– In the abstract, I would suggest that immunotherapeutic applications of design approaches are "underexplored" rather than "unexplored".

– In the introduction, I believe that it is not fully accurate to say that ProteinMPNN designs “entirely novel proteins”, as the backbone is not novel (fixed). Perhaps this could be amended to “novel sequences”.

– In section 2.1, a more likely explanation for the poorer performance on MHC-II- than MHC-I-binding TCRs is the lower number of training structures, not test structures. There does not seem to be any evidence to support that the test cases included here would be more challenging than other potential MHC-II test cases.

– Towards the end of section 2.1, “interestingly” should be removed. As the authors point out, sequence recovery is expected to be higher across all rather than just interface CDR3 positions due to greater sequence conservation.

– The last sentence of section 2.4 would require more supporting evidence. The data could alternatively be interpreted as the sequence recovery remaining unchanged due to the relatively low/moderate recovery success; the models could, for example, only be recovering non-interface or non-essential binding positions.

• The ESM-IF1 inverse folding model should be referred to as “ESM-IF1” rather than “ESM-IF”.

• The figure colors should be kept consistent throughout (e.g., always green for native).

• In Figure 2B, one CDR3b appears to be in a very different place than the other CDRs. Could the authors comment on this?

• The amino acid groupings in section 4.4 are missing P and T. Additionally, it would be helpful to keep the groupings consistent with Figure 3.

• Figure S12B should be square (equal x- and y-axes) to show that, although the correlation is strong, the calculated and experimental dG results are not on exactly the same scale.

• The caption of Figure S4 only explains A-B, but the figure includes panels A-D.

**Have the authors made all data and (if applicable) computational code underlying the findings in their manuscript fully available?**

Reviewer #1: Yes

Reviewer #2: Yes

PLOS authors have the option to publish the peer review history of their article (what does this mean?). If published, this will include your full peer review and any attached files.

Reviewer #1: No

Reviewer #2: **Yes: **Alissa M Hummer
---

## [Decision Letter · Decision Letter 1]

26 Aug 2024

Dear Dr Ribeiro,

Thank you very much for submitting your manuscript "Exploring the Potential of Structure-Based Deep Learning Approaches for T cell Receptor Design" for consideration at PLOS Computational Biology. As with all papers reviewed by the journal, your manuscript was reviewed by members of the editorial board and by several independent reviewers. The reviewers appreciated the attention to an important topic. Based on the reviews, we are likely to accept this manuscript for publication, providing that you modify the manuscript according to the review recommendations.

Sincerely,

Dina Schneidman

Academic Editor

PLOS Computational Biology

Nir Ben-Tal

Section Editor

PLOS Computational Biology

Reviewer's Responses to Questions

**Comments to the Authors:**

Reviewer #1: The authors have addressed all of my comments.

Reviewer #2: The authors have made significant revisions to the manuscript, addressing the feedback from the reviewer comments and strengthening the paper. The added analyses provide a clearer understanding of the strengths, limitations and functionalities of inverse folding models for TCR sequence design. Additionally, the updates to the Discussion section highlight interesting points and areas for future exploration.

I have a few minor comments arising from the changes. The line numbers are indicated for the version with tracked changes.

Minor analysis

• Line 171, Fig S7: Are there any unifying features for the cases which ESM-IF1 or ProteinMPNN, respectively, performed better on? (i.e. those which are above vs. below y=x in Fig S7)

• Fig 5: Could you include a figure and statistical analysis similar to Fig 5B for the information portrayed in Fig 5A? (i.e. Fig 5B but with 'maximum sequence recovery' rather than 'maximum sequence recovery difference')

• Line 512, Fig S27: Can you identify any explanations (e.g., specific sequence changes) for why designing all CDRs substantially improved binding affinity as compared to designing only CDR3 for 7pbe?

Text – clarification

• Line 352: Could this new text be rephrased for clarity? The statements in lines 349 and 352 seem contradictory at a first glance (the differences being analyzed in these sentences – with vs. without the pMHC; ProteinMPNN vs. ESM-IF1 – could be explained more clearly). The sentences in lines 354-357 also appear contradictory; these concepts are more clearly explained in the response to reviewers comment.

Text – minor changes

• Line 24: The phrase "trained on" in "this study explores whether computational methods, trained on deep learning architectures such as ProteinMPNN and ESM-IF1" is misleading and should be amended to clarify that ProteinMPNN and ESM-IF1 are directly used (as opposed to new methods trained/based on these architectures).

• Line 336: It would be helpful if more information was directly provided in the Results section about the method for identifying hotspots (using Rosetta, 0.5 kcal/mol cutoff) and the relevant methods section (4.7) was referenced.

• A few typos are included and should be fixed, e.g., "residues" to "resides" in line 39 and "10-10" to "10^-10" in line 296.

Figures – minor changes

• Figure S3C – The sequence position numbers are absent from panel C (included in A, B and D).

**Have the authors made all data and (if applicable) computational code underlying the findings in their manuscript fully available?**

Reviewer #1: **No: **

Reviewer #2: None

PLOS authors have the option to publish the peer review history of their article (what does this mean?). If published, this will include your full peer review and any attached files.

Reviewer #1: No

Reviewer #2: **Yes: **Alissa M. Hummer

Figure Files:

Data Requirements:

Reproducibility:

References:

---

## [Editor Report · Decision Letter 2]

14 Sep 2024

Dear Dr Ribeiro,

We are pleased to inform you that your manuscript 'Exploring the Potential of Structure-Based Deep Learning Approaches for T cell Receptor Design' has been provisionally accepted for publication in PLOS Computational Biology.

Best regards,

Dina Schneidman

Academic Editor

PLOS Computational Biology

Nir Ben-Tal

Section Editor

PLOS Computational Biology

---

## [Editor Report · Acceptance letter]

24 Sep 2024

PCOMPBIOL-D-24-00822R2 

Exploring the Potential of Structure-Based Deep Learning Approaches for T cell Receptor Design

Dear Dr Ribeiro,

I am pleased to inform you that your manuscript has been formally accepted for publication in PLOS Computational Biology. Your manuscript is now with our production department and you will be notified of the publication date in due course.

With kind regards,

Anita Estes
